

# Quantifying Shallow Subsurface Water and Heat Dynamics using Coupled Hydrological-Thermal-Geophysical Inversion

Anh Phuong Tran[1], Baptiste Dafflon[1], Michael B. Kowalsky[1], Philip Long[1], Tetsu K. Tokunaga[1], Kenneth H. Williams [1], Susan S. Hubbard[1],

[1]Climate & Ecosystems Division, Earth and Environmental Sciences Area, Lawrence National Berkeley Lab, Berkeley, California, CA 94720, USA

*Correspondence to*: Anh Phuong Tran (aptran@lbl.gov)

**Abstract.** Improving our ability to estimate the parameters that control water and heat fluxes in the shallow subsurface is particularly important due to their strong control on recharge, evaporation, and

biogeochemical processes. The objectives of this study are to develop and test a new inversion scheme to simultaneously estimate subsurface hydrological, thermal and petrophysical parameters using hydrological, thermal and electrical resistivity tomography (ERT) data. The inversion scheme, which is based on a nonisothermal, multiphase hydrological model, provides the desired subsurface property estimates in high spatiotemporal resolution. A particularly novel aspect of the inversion scheme is the

explicit incorporation of the dependence of the subsurface electrical resistivity on both moisture and temperature. The scheme was applied to synthetic case studies, as well as to real datasets that were autonomously collected at a biogeochemical field study site in Rifle, Colorado. At the Rifle site, the coupled, hydrological-thermal-geophysical inversion approach reproduced a relatively accurate time-series of measured matric potential, temperature, and apparent resistivity. Synthetic studies found that

neglecting the subsurface temperature variability, and its effect on the electrical resistivity in the hydrogeophysical inversion, may lead to an incorrect estimation of the hydrological parameters. The approach is expected to be especially useful for the increasing number of studies that are taking advantage of autonomously collected ERT and soil measurements to explore complex terrestrial system dynamics.



# 1 Introduction

Shallow subsurface moisture and temperature are two primary variables that play key roles in hydrological and biogeochemical processes in terrestrial environments. For example, watershed moisture content and temperature control the partitioning of precipitation into evapotranspiration,

infiltration and runoff (Merz and Bardossy, 1998; Brocca et al., 2010). Ecosystem moisture content and temperature conditions are closely linked to form, functioning and organization of vegetation, which in turn influences ecological diversity (Rodriguez-Iturbe, 2000). Subsurface moisture and temperature largely influence microbial activity in the subsurface, including respiration of greenhouse gases (Boone et al., 1998; Luo et al., 2013). However, monitoring the variability of subsurface moisture

and temperature over spatial temporal scales that are relevant to the native processes yet informative for predicting watershed or ecosystem functioning is challenging. Conventional point sensing approaches can provide direct measurements of subsurface moisture and temperature. However, due to labour and cost involved in installing point sensing systems and the invasive nature of the sensors, the spatial support scale of point sensing systems is typically quite small compared to the scale of system of

interest.

Over the last two decades, many hydrogeophysical approaches have been developed to combine point and geophysical measurements for improved subsurface property estimation or process monitoring (see reviews provided by Rubin and Hubbard, 2005 and Hubbard and Linde, 2011; Binley et al., 2015).

Statistical approaches have been extensively used to integrate point measurements with commonly collected geophysical datasets, such as Ground Penetrating Radar (GPR) and Electrical Resistance Tomography (ERT). For example, Hubbard et al. (2001) applied a Bayesian algorithm to integrate surface and crosshole GPR, seismic crosshole tomography, cone penetrometer, and borehole electromagnetic flowmeter to estimate the spatial distribution of subsurface hydraulic conductivity.

Binley et al. (2002) estimated shallow subsurface hydraulic conductivity using both cross-well ERT and GPR. Doetsch et al. (2010) showed that merging seismic, GPR and ERT data could significantly improve the accuracy of aquifer zonation and associated zonal parameter estimation. Dafflon and Barrash (2012) used a stochastic approach to estimate the distribution of porosity from well data and





ground-penetrating radar (GPR) data. Tran et al. (2015) combined surface GPR and frequency domain reflectometry data to better quantify the spatiotemporal dynamics of moisture along a hill slope. Coupled hydrogeophysical inversion approaches have also been developed to improve quantification of subsurface processes, which assimilate all geophysical and other key datasets into a models that

consider physical hydrodynamics (i.e., Darcy's law) and electromagnetic laws (i.e., Maxwell's equations). Because coupled inversion approaches permit direct use of geophysical data for inversion, it avoids the errors typically associated with geophysical inversion process (e.g., Binley et al., 2002; Singha and Gorelick, 2005) and associated resolution issues (Day-Lewis and Lane, 2004). Kowalsky et al. (2005) and Lambot et al. (2009) developed coupled inversion schemes, and used time-lapse GPR

data to estimate hydraulic conductivity and matric potential functions. Johnson et al. (2009) jointly inverted time-lapse hydrogeologic and ERT data without a priori assumptions about petrophysical parameters. Using ERT data, Huisman et al. (2010) developed a coupled Bayesian hydrogeophysical inversion approach to determine the hydraulic properties and its uncertainties of flood-protection dikes. Kowalsky et al. (2011) employed time-lapse ERT, groundwater level, and nitrate concentration data to

estimate hydrogeochemical parameters and behaviour of a contaminated subsurface system.

Recent hydrogeophysical studies have illuminated several challenges as well as opportunities (e.g., Binley et al, 2015). For example, current integration approaches still suffer from difficulties inherent to geophysical inversion, such as the need for user-determined regularization, or error in mass balance. It

is still challenging to account for the varying spatial and temporal resolution of the different measurement techniques, and to measure subsurface hydrological and thermal variables with high-enough spatial and temporal resolution to capture the dynamics of these variables over relevant field scales. On the other hand, the emergence of autonomous acquisition platforms is opening new doors for characterizing and monitoring terrestrial systems in high resolution. Autonomous data acquisition

paired with increasing computational capabilities provides new opportunities to rapidly assimilate field data into predictive models, potentially facilitating quantification of terrestrial system behaviour and moving toward real-time management of terrestrial system resources.



To date, ERT is the geophysical approach that is most commonly collected in an autonomous manner. ERT provides information about the distribution of subsurface electrical resistance; a review of ERT theory and inversion procedures is given by Binley and Kemna (2005). Due to the typically high sensitivity of electrical resistivity to pore fluid conductivity and saturation, ERT has been used widely

for monitoring the vadose zone soil moisture and other terrestrial system processes (e.g., Binley et al., 2002; Kemna et al., 2002; McClymont et al., 2013; Hubbard et al., 2013). However, because the electrical resistivity is also sensitive to other subsurface properties (such as porosity, tortuosity, pore-grain electrochemistry, mineralogy and temperature) other measurements must be used with ERT to avoid large estimation errors (Binley et al., 2002). For example, dependence of subsurface electrical

resistivity on temperature is well known but often not adequately accounted for in hydrogeophysical approaches. The subsurface temperature directly influences the subsurface electrical resistivity. It also controls the phase change of subsurface moisture, and ultimately affects the subsurface resistivity. In some cases, subsurface temperature variations affect subsurface resistivity more than moisture variations (Rein et al., 2004; Musgrave and Binley, 2011). The conventional approach for correcting for

temperature effects on ERT data includes inverting the data and then performing correction on the obtained resistivity/conductivity image (Hayley et al., 2007; Ma et al., 2014). This approach is not suitable for the coupled hydrogeophysical inversion, because the objective of the hydrogeophysical inversion is to estimate hydrological parameters (not electrical resistivity/conductivity image). Hayley et al. (2010) proposed a temperature-compensation approach that removes the temperature effect on the

data before inversion, which appeared to better resolve the temperature dependence of the electrical resistivity. This approach can be used for the hydrogeophysical inversion. However, this approach first requires the inversion of electrical resistance data to obtain the correction factors. Secondly, the correction usually relies on temperature measurements at several specific points in time, which may not suffice due to high variability of moisture and temperature in space and time. To date, no studies have

incorporated and evaluated the effect of the relationship between subsurface resistivity and temperature into a coupled hydrogeophysical inversion scheme.

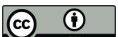



The opportunities and challenges identified above motivate the three key objectives of this study: to (1) develop a coupled hydrological-thermal-geophysical inversion scheme that is capable of incorporating non-isothermal behaviour of the shallow subsurface as well as multiphase of moisture into hydrogeophysical inversion, and that jointly uses different thermal, hydrological and geophysical data

for inversion including ERT; (2) apply the developed inversion scheme to estimate hydrological (permeability and van Genuchten curve parameters), thermal (heat conductivity) and petrophysical (Archie's model parameters) parameters to assess the evaporation/infiltration processes at a Department of Energy (DOE) experimental field site in Rifle, Colorado; and (3) perform synthetic studies to explore the importance of consideration of subsurface temperature variability and its direct and indirect

influence on the electrical resistivity in the hydrogeophysical inversion. To our knowledge, this is the first study that explicitly integrates both direct and indirect dependence of electrical resistivity on temperature in the coupled hydrogeophysical inversion. While tested at the Rifle CO site, we envision the new inversion approach to be widely useful at other study sites, particularly those that can take advantage of autonomous ERT and other datasets.

We organize this article as follows. Section 2 describes the development of the hydrological-thermal-geophysical inversion scheme. The application of the inversion scheme to the Rifle site study is described in Section 3. Section 4 compares two synthetic cases that perform geophysical inversion, with and without considering the subsurface temperature's influence. Section 5 offers summary and

concluding remarks.

## 2 Methodology

### 2.1 Hydrological forward model

In this study, we simulated the nonisothermal two-phase (gas and liquid), three-component (air, water and heat) flow in the vadose zone using the integral finite-difference simulator TOUGH2 (Pruess et al.,

1999). TOUGH2 solves the mass and energy balance equations for each component over an arbitrary



volume $V$, confined by a closed surface $\Gamma$ of the flow computational domain, which is written in the integral form as below:

$$\int_V M^k dv = \int_\Gamma F^k n d\Gamma + \int_V q^k n dv \tag{1}$$

in which $M$ is the mass or energy accumulation term for component $k$; $F$ represents the mass or heat

flux; $q$ denotes the sink or source terms; and $n$ is the normal vector on the surface element d$\Gamma$. The mass accumulation term is defined as:

$$M^k = \phi \sum_\beta S_\beta \rho_\beta X_\beta^k \tag{2}$$

where $\varphi$ is the porosity; $S_\beta$ and $\rho_\beta$ are, respectively, the saturation and density of phase $\beta$; $X_\beta^k$ is the mass fraction of phase $\beta$ in component $k$. For simulating the nonisothermal problem, the heat

accumulation component ($M^h$) is also accounted for:

$$M^h = (1 - \phi)\rho_R C_R T + \phi \sum_\beta S_\beta \rho_\beta u_\beta \tag{3}$$

where $\rho_R$ and $C_R$ are, respectively, the grain density and specific heat capacity of the soil/sediment particle materials; $T$ is the temperature; and $u_\beta$ is the specific internal energy in phase $\beta$.

The mass flux term $F^k$ of component $k$ is the sum of all of its phase fluxes:

$$F^k = \sum_\beta f_\beta X_\beta^k \tag{4}$$

with

$$f_\beta = -K \frac{k_{r\beta}}{\mu_\beta} \rho_\beta (\nabla P_\beta - \rho_\beta g) - \phi S_\beta^k d_\beta^k \nabla X_\beta^k \tag{5}$$

The first term in Equation (5) represents the advection. The second one describes the molecular diffusion. K is the absolute permeability; $k_{r\beta}$ is the relative permeability of phase $\beta$; $P_\beta = P_{ref} + P_c$

denotes the pressure in which $P_{ref}$ and $P_c$ are the reference gas pressure and matric potential; $d_\beta^k$ and $\mu_\beta$ are the molecular diffusion coefficient and viscosity of phase $\beta$; and $g$ is the gravitational acceleration. For gas phase (air and vapor), the diffusion coefficient is a function of pressure and temperature as:



$$d_g^k = d_g^k(P_0, T_0) \ \frac{P_0}{P} \left[ \frac{T+273.15}{273.15} \right]^{1.8} \tag{6}$$

where $d_g^k(P_0, T_0)$ is the gas diffusion coefficient at the standard condition $P_0 = 1$ atm, and $T_0 = 0^0$C. The relationship between the matric potential, the relative permeability, and the water saturation is formulated by Mulem-van Genuchten (van Genuchten, 1980) as:

$$P_c = -\frac{1}{\alpha} \left( S_e^{-1/m} - 1 \right)^{1-m} \tag{7}$$

$$k_{rl} = \sqrt{S_e} \left[ 1 - \left( 1 - S_e^{1/m} \right)^m \right]^2 \tag{8}$$

The relative permeability of the gas phase is described by Corey (1954) as:

$$k_{rg} = \begin{cases} (1-\hat{S})^2(1-\hat{S}^2) & if \quad S_{gr} > 0 \\ 1 - k_{rl} & if \quad S_{gr} = 0 \end{cases} \tag{9}$$

where $S_e$ and $\hat{S}$ are defined as:

$$S_e = \frac{S_l - S_{lr}}{S_{ls} - S_{lr}} \tag{10}$$

$$\hat{S} = \frac{S_l - S_{lr}}{1 - S_{lr} - S_{gr}} \tag{11}$$

in which $S_{ls}$ represents the saturated liquid saturation; $S_{lr}$ and $S_{gr}$ are the residual liquid and gas saturation, respectively; $m$ represents the pore size distribution of the soil/sediment; and α is inversely proportional to the air-entry pressure. Heat fluxes consist of conductive and convective components:

$$F^h = -\lambda \nabla T + \sum_\beta h_\beta f_\beta \tag{12}$$

where $\lambda$ demotes the heat conductivity, and $h_\beta$ is the specific enthalpy in phase $\beta$. It is worth noting that the evaporation was accounted for by the diffusion term in Equation (5). However, the current version of TOUGH2 does not consider the root water uptake and transpiration from vegetation.



## 2.2 ERT forward model

The ERT forward model solves Poisson's equation, which describes the relationship between the potential field due to a given input current and the electrical conductivity distribution. In this study, we used the forward model of the Boundless Electrical Resistivity Tomography (BERT) package, developed by Rucker et al. (2006). BERT numerically solves Poisson's equation using the finite element method in a three-dimensional, arbitrary topography. By incorporating unstructured, tetrahedral meshes, the model enables efficient refinement of the local mesh, and flexibly describes any geometry of the computational domain. The use of quadratic shape functions also helps to improve the accuracy of the simulation.

## 2.3 Petrophysical model

The bulk electrical conductivity ($\sigma_a$) includes contributions from the electrical conductivity of pore water and the surface conduction at the pore and water-mineral interface (Revil et al., 2012). We employed the model proposed by Linde et al. (2006), which was extended from Archie's model (Archie, 1942), and is expressed below as:

$$\sigma_a = \phi^d [S_l^n \sigma_w + (\phi^{-d} - 1)\sigma_s] \tag{13}$$

where $d$ is the cementation index; $n$ is the saturation index; and $\sigma_w$ and $\sigma_s$ are, respectively, the electrical conductivity of water and soil/sediment surface conduction. Equation (13) indicates that the cementation and saturation indexes closely correlate. Hence, to reduce the number of unknown parameters and ameliorate nonuniqueness, we set the cementation index at $d = 1.3$, which is commonly used for unconsolidated sand (Archie, 1942). The electrical conductivity of water, which does not vary significantly over time at the Rifle site, was taken from the measurements at the nearby well, and is equal to 0.244 S/m.

The relationship between temperature and electrical conductivity can be formulated using linear (Sen and Goode, 1992) or exponential (Llera et al., 1990) equations. In this study, we chose the linear form:

$$\sigma_a^T = \sigma_a [1 + c(T - 25)] \tag{14}$$





in which $T$ is the temperature; $\sigma_a^T$ is the electrical resistivity at temperature $T$°C; and $c$ is the temperature compensation factor corresponding to 25°C. The value $c = 0.0183$°C$^{-1}$ as suggested by Hayley et al. (2007) was used for our study. The electrical resistivity is the reciprocal of the electrical conductivity $\rho_a^T = 1/\sigma_a^T$.

## 2.4. Coupled hydrological-thermal-geophysical inversion scheme

We developed the coupled hydrological-thermal-geophysical inversion scheme within iTOUGH2 (see Finsterle (1999) and Finsterle et al. (2012)). Figure 1 presents the flowchart of the scheme, which includes seven steps: (1) simulate subsurface moisture content and temperature using the TOUGH2 model; (2) transform the simulated moisture content to an electrical conductivity image using petrophysical relationships; (3) apply the temperature correction for the electrical conductivity using the simulated temperature, and convert the corrected conductivity to a resistivity image; (4) interpolate the electrical resistivity image from the TOUGH2 computational mesh to the BERT mesh; (5) execute the forward BERT model to simulate the electrical resistance from the resistivity image; (6) convert the electrical resistance to the apparent resistivity using geometric factors; (7) minimize the misfit between simulation and measurement of the apparent resistivity and other hydrological-thermal (matric potential and temperature) data to estimate hydrological-thermal-petrophysical parameters. The misfit is formulated by the objective function as below:

$$\Phi(p) = e^T C^{-1} e \tag{15}$$

where $e = z^* - z(\theta, p)$ is the residual vector quantifying the difference between the modeled ($z$) and measured ($z^*$) data; $p$ and $\theta$ are, respectively, the vectors representing the model parameters and input data; and $C$ denotes the covariance matrix of measurements errors. We assumed that there is no correlation between measurement errors, and therefore the covariance matrix $C$ becomes a diagonal matrix in which the main diagonal elements are the variances of measurement errors. It is worth noting that the vectors $z$, $z^*$ and matrix $C$ can contain multiple data types. The difference in units associated with the different data types was removed by the covariance matrix of the measurement errors. For parameter estimation, we used the Levenberg-Marquardt algorithm (Marquardt, 1963) for nonlinear optimization.





The agreement between measured and modeled data was evaluated using the Nash-Sutcliffe efficiency coefficient:

$$NSE = 1 - \frac{e^T e}{n_0 \sigma_0^2} \tag{16}$$

5    where $\sigma_0^2$ is the variance of the measured data, and $n_0$ is the number of measurements. The Nash-Sutcliffe coefficient ranges from $-\infty$ to 1. The modeled and measured data perfectly agree if this coefficient equals 1. A coefficient of 0 implies that the model prediction is as accurate as the mean of the measured data; a value less than zero indicates that the model prediction is worse than the measured mean.

The uncertainties of estimated parameters are characterized by their standard deviation values, which are the square root of the diagonal elements of the covariance matrix of the estimated parameters:

$$\sigma_{p_i} = \sqrt{C_{pp_{ii}}} \tag{17}$$

where $i = 1, ..., n_p$. The covariance matrix of the estimated parameters is computed as:

15    $$C_{pp} = s^2 (J^T C^{-1} J)^{-1} \tag{18}$$

where $J$ is the Jacobian matrix, and $s^2$ is an estimate of the error variance:

$$s^2 = \frac{e^T C^{-1} e}{n_0 - n_p} \tag{19}$$

Good initial guesses can help to avoid local minima with unrealistic solutions. As such, we implement the following practical procedure to progressively approach an optimal solution:

20    1. Invert the matric potential data to obtain the subsurface hydrological parameters. In this step, we consider only the one-dimensional isothermal hydrological model.

    2. Use the subsurface temperature data to estimate the thermal parameters of the one-dimensional nonisothermal hydrological model. The subsurface hydrological parameters obtained in step 1 are fixed, and are used to simulate the hydrological processes.





3. Jointly invert the matric potential, temperature and apparent resistivity data to obtain the subsurface hydrological-thermal and petrophysical parameters. The hydrological-thermal parameters from steps 1 and 2 are used as the initial guesses for this step. In this step, the inversion is performed for the two-dimensional nonisothermal hydrological model.

In each step, global sensitivity analysis is performed to evaluate the sensitivity of the calibration data with respect to the model parameters. The insensitive parameters (the sensitivity coefficient is approximately equal to zero) are not considered in inversion. We apply the global sensitivity analysis method referred to as one-step-at-a-time (OAT) Morris method, which is available in iTOUGH2

10 (Wainwright et al., 2013). This method is briefly described as follows: the parameter space of $n_p$ parameters is normalized to $n_p$−dimensional domain ($[0, 1]^{n_p}$). Each dimension of this normalized domain is discretized into $n_s$−1 equal segments, generating $n_s$ grid points that take values in the set $\{0, 1/(n_s − 1), 2/(n_s − 1), ...1\}$. The element effect ($EE^j(p_i)$) of parameter $p_i$ at an arbitrary grid point with respect to model output z is defined as:

$$EE^j(p_i) = \frac{1}{F} \frac{z\left(p_1^j, ..., p_i^j + \Delta, ..., p_{n_p}^j\right) - z\left(p_1^j, ..., p_i^j, ..., p_{n_p}^j\right)}{\Delta} \qquad (20)$$

in which $p^j \leq 1 − \Delta$, with $\Delta = \frac{n_s}{2(n_s−1)}$, and $F$ is the scaling factor for comparing the element effects of different measurements $z$. The element effect quantifies the variation of the model output with respect to variation of parameter $p_i$ at a given point in the parameter space. To evaluate the parameter sensitivity, we need to calculate the element effects of all parameters at all grid points, which requires a large

20 computing resource. To overcome this constraint, Morris (1991) generated several random sample paths, and computed the element effects of each parameter along these paths. The sensitivity coefficient $|EE(p_i)| = \frac{1}{n_s} \sum_{j=1}^{j=n_s} |EE^j(p_i)|$ determines the sensitivity of parameter $p_i$. A parameter with a higher sensitivity coefficient is more sensitive than the other parameters.



## 3 Field study

### 3.1 Study site and datasets

The newly developed approach was tested at the within floodplain adjoining the Colorado River, near Rifle, Colorado (USA) (Figure 2). The perched aquifer at the site overlies low permeability mud and siltstones of the Eocene Wasatch Formation. Above the Wasatch Formation is a Quaternary alluvial layer consisting of sandy gravelly unconsolidated sediments. The uppermost layer is a silty clay fill with a thickness of around 1.5–2 m, which replaced contaminated soils and sediments removed from the site following uranium reclamation activities. Groundwater elevations fluctuate seasonally with snowmelt infiltration and Colorado River stage, and vary from around 3.5 to 2.4 m below ground surface.

The Berkeley Lab and others in the scientific community have performed many studies at the Rifle site to explore complex subsurface hydro-biogeochemical behaviour and to test the development of new characterization and modelling approaches. For example, Li et al. (2010) used reactive transport modeling to investigate the influence of physical and geochemical heterogeneities on the spatiotemporal distribution of mineral precipitates and biomass that formed during a biostimulation experiment. Yabusaki et al. (2011) developed a three-dimensional hydro-biogeochemical reactive transport model of Rifle to improve understanding of the uranium variability, hydrological conditions, and soil properties, under the pulsed acetate amendment, Chen et al. (2013) developed a data-driven biogeophysical approach to quantify redox-driven biogeochemical transformations using geochemical measurements and induced polarization data. Wainwright et al (2015) used induced polarization data and stochastic methods to estimate the spatial distribution of naturally reduced zones in the subsurface, which served as biogeochemical hotspots; the geophysical-obtained information was used to constrain simulations of biogeochemical cycles across the Rifle floodplain. Arora et al. (2016) used reactive transport modelling approaches to explore seasonal variations in biogeochemical fluxes occurring from bedrock to canopy as well as laterally to the Colorado River. They found that $CO_2$ concentration in the unsaturated zone could not be accurately reproduced without incorporating temperature gradients in the simulations, and that incorporating temperature fluctuations resulted in an increase in the annual groundwater carbon fluxes to the river by 170%. They concluded that spatial microbial and redox



zonation as well as temporal fluctuations of temperature and water table depth contributed significantly to subsurface carbon fluxes in the Rifle floodplain, and identified the need to represent temperature and moisture dynamics for accurate model simulations.

In this study, we tested our new approach using data collected along a Rifle CO ERT transect, which includes 112 electrodes with a distance between any two adjacent electrodes of 1 m (Figure 2). The ERT data were autonomously collected every day from April through June 2013 using the Wenner electrode array. These data were used for two purposes: (1) determining subsurface stratigraphy to support construction of the hydrological model; (2) estimating hydrological-thermal and petrophysical

parameters through the coupled inversion approach.

For characterizing subsurface stratigraphy and specifying the depths of the fill, alluvium and Wasatch layers, we used the BERT inversion package (Gunther et al., 2006) to invert the ERT data that were collected on May 20, 2013. The electrical resistivity image obtained by inversion is shown in Figure 3.

The three layers are clearly visible in the figure, where the resistivity values of the fill and Wasatch layers are less than those of the alluvium. This is a plausible result, as the fill and Wasatch layers contain higher silt/clay content, and therefore they conduct more than the alluvium layer. We developed a computational domain that is a rectangle centered at the TT02 well, with a width of 30 m, as shown in Figure 3b. Previous work at this site has suggested that the spatial variability over the extent of the

simulation transect is not likely to be significant (Li et al., 2010). Consequently, we assumed the computational domain includes two homogeneous layers: namely, fill and alluvium. The porosity is 0.4 for the fill, and 0.2 for the alluvium layer. The top boundary of the domain is the atmospheric layer, and the bottom is the impermeable Wasatch layer. We set the depth of the bottom boundary at the average depth of the Wasatch layer, $z = 6.5$ m. The domain was divided into 30 equally spaced columns, each

with a size of 1 m in the horizontal direction. In the vertical direction, the cell size is 0.05 m for the uppermost 2 m, 0.3 m for the next 1.5 m, and 0.6 m for the last 3 m, for a total of 1560 cells.




We performed the hydrological-thermal simulation during the snow-free period from May 04, 2013, to November 25, 2013 (194 days). All of meteorological data (atmospheric pressure, temperature, humidity, and rainfall) were measured at a nearby meteorological station. The surface boundary conditions include land surface temperature, atmospheric pressure, air mass fraction, and rainfall. The

land surface temperature was adjusted from the atmospheric temperature, based on a regression approach proposed by Zheng et al. (1993), while the air mass fraction was calculated from the atmospheric pressure and relative humidity data. The bottom boundary condition of pressure was calculated from the groundwater table data, and the bottom temperature was approximated from the land surface temperature. The initial conditions were derived from the measured data at the beginning

of the simulation period. For more detailed information about initial and boundary conditions, we refer to Tran et al. (2016).

Data for inversion included time-lapse matric potential, temperature, and resistivity measurements. The temperature was measured every 5 minutes at six depths below the surface at the TT03 well: $z = 0.75$, 1,

1.5, 2.5, 4.6 and 6 m (see the TT03 location in Figure 2). The 5-minute data were averaged to obtain daily data. Using tensiometers, the matric potential was occasionally measured at the TTO2 well at depths $z = 0.5$, 1, 1.5, 2, 2.5 and 3 m. As for the ERT data, we chose six datasets that cover the most important variations of subsurface moisture and temperature during the measurement period. For each dataset, we selected 246 values obtained from 54 electrodes in and around the computational domain

for inversion. The measurement errors were assumed to follow a standard Gaussian distribution. The standard deviation of the errors for the resistivity and matric potential data are 5% of the measurement values. For the temperature data, because the instrument errors of the thermistors are from 0.1 to 0.4°C, we assumed that the standard deviation of the errors for this type of measurement is 0.4°C.

### 3.2 Results and Discussion

All of the hydrological-thermal and petrophysical parameters that were considered in this study are presented in Table 1. The first and second columns present the parameter names and ranges, respectively. From the third to the last column, we present the estimated parameters obtained from

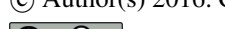



different inversion cases, namely hydrological inversion (HI), thermal inversion (TI), and coupled hydrological-thermal-geophysical inversion (HTGI).

### 3.2.1 Sensitivity analysis

The sensitivity coefficient of the matric potential data with respect to the subsurface hydrological
parameters at all measured depths (0.5, 1, 1.5, 2, 2.5, 3 m) is shown in Figure 4a. The figure indicates that the matric potentials are more sensitive to the parameters of van Genuchten's retention curve than to the absolute permeability. At the fill layer, the influence of parameter α on the matric potential is significantly higher than the other parameters. The second most sensitive parameter is $m$. At the alluvium layer, α and $m$ are also the two most sensitive parameters. The sensitivity coefficient of the
absolute permeability of the fill layer ($|EE|_{K_{fill}}^{matric\ potential}$) is relatively small, and that of the alluvium permeability is nearly equal to zero. This can be explained by the lack of infiltration during the simulation period. As a result, there is little information for estimating the absolute permeability, which controls the moisture dynamics. The retention curve parameters determine the shape of the matric potential profile, and therefore the matric potential is more sensitive to them. Figure 4a also shows that
the hydrological parameters of a given layer are mostly sensitive to the matric potential measurements at that layer. For example, the $|EE|_{\alpha_{fill}}^{matric\ potential}$ of α of the fill layer on the matric potential is around 45–68 for the fill layer, and zero for the alluvium layer ($z > 1.5$ m). By contrast, the $|EE|_{\alpha_{fill}}^{matric\ potential}$ of α of the alluvium layer on the matric potential is 35–114 for the alluvium layer, and zero for the fill layer. This implies that there was little moisture exchange between the two layers during the simulation
period.

### 3.2.1.2 Sensitivity of temperature with thermal parameters

The sensitivity of the subsurface temperature data with respect to the heat conductivity of the fill and alluvium layers at depths from 0.75 to 6 m is depicted in Figure 4b. The figure indicates that the sensitivity coefficient $|EE|_{\lambda_{fill}}^{temperature}$ of the heat conductivity of the fill layer on temperature reaches
its maximum at $z = 1.5$ m ($|EE|_{\lambda_{fill}}^{temperature} = 6.5$) and its minimum at $z = 6$ m ($|EE|_{\lambda_{fill}}^{temperature} = 0.2$).



This implies that the temperature data at 1.5 m depth contain the most valuable information for estimating the heat conductivity of the fill layer. The temperature data at depth $z = 4.6$ m are the most sensitive to the heat conductivity of the alluvium layer $|EE|_{\lambda_{alluvium}}^{temperature} = 2.3$, while the temperature data at depths 0.75, 2.5 and 6 m are the least sensitive, with the $|EE|_{\lambda_{alluvium}}^{temperature}$ roughly equal to 0.7.

The figure also indicates that the subsurface temperature data at depths 0.75, 1, 1.5 and 2.5 m are much more sensitive to the heat conductivity of the fill than to that of the alluvium layer. By contrast, below 2.5 m, the sensitivity of the temperature data to the heat conductivity of the fill layer is slightly higher than to that of the fill layer. This is because temperature at shallower depths is more dynamic in both time and space than at deeper depths. As a result, there is more information for estimating the heat

conductivity of the shallower, fill layer.

### 3.2.1.3 Sensitivity of apparent resistivity with hydrological-thermal and petrophysical parameters

Based on the above sensitivity analysis with the matric potential and temperature data, we selected the six most sensitive hydrological-thermal parameters (α, $m$ and λ, for both fill and alluvium layers) for sensitivity analysis with the apparent resistivity data. We also considered three petrophysical

parameters, including $n$ (fill, alluvium), and $\sigma_s$ (fill) (Equation 13). Figure 4c presents the sensitivity coefficient $|EE|$ of the six apparent resistivity datasets collected at different dates with respect to nine parameters. The figure shows that the apparent resistivity data are much more sensitive to the petrophysical parameters than to the hydrological-thermal ones. Of the petrophysical parameters, the saturation index ($n$) of the alluvium layer is the most sensitive parameter. Among all hydro-thermal

parameters, apparent resistivity is the most sensitive to the heat conductivity of the fill layer, implying that for this study the influence of temperature on electrical resistivity is larger than that of moisture content. The apparent resistivity data are more sensitive to the hydrological-thermal parameters of the fill layer than those of the alluvium layer. The $|EE|$ of all hydrological-thermal parameters of the alluvium layer is mostly equal to zero. This is because moisture and temperature exhibit larger

variations in the fill than in the alluvium layer.



### 3.2.2 Inversion results

The estimated parameters and their associated uncertainties based on hydrological, thermal and coupled hydrological-thermal-geophysical inversions are presented in Table 1. For the hydrological inversion, we used the matric potential data to estimate six hydrological parameters: $\alpha$ (both fill, alluvium), $m$

(both fill, alluvium), $K$ (fill), and $D(P_0,T_0)$. Because the matric potential data are negligibly sensitive with the permeability of the alluvium layer ($K$ (alluvium)), we did not consider $K$ (alluvium) in hydrological inversion. We set it at $7.95 \times 10^{-12}$ m$^2$, which is the value averaged from well measurements. For the thermal inversion, we estimated the heat conductivity ($\lambda$) of the fill and alluvium layers using temperature data. The specific heat capacity of the soil/sediment particles of both fill and

alluvium layers was fixed at their typical value, $C_R = 870$ Jkg$^{-1}$C$^{-1}$ (Campbell and Norman, 1998). For the coupled hydrological-thermal-geophysical inversion, we estimated six parameters including three hydrological-thermal parameters, $m$ (fill), $\alpha$ (fill) and $\lambda$ (fill), and three petrophysical parameters, $n$ (both fill, alluvium) and $\sigma_s$ (fill), using the matric potential, temperature and apparent resistivity data. The surface conduction of the alluvium layer was set to zero. Because the apparent resistivity data show

little sensitivity to the hydrological-thermal parameters of the alluvium layer, these parameters are not improved by the coupled inversion. Therefore, they were fixed at the values obtained from the hydrological and thermal inversion. The initial guesses for the fill hydrological-thermal parameters were obtained from the previous hydrological and thermal inversion.

The hydrological inversion reveals that, compared to the other hydrological parameters, the uncertainty of the absolute permeability ($K$) of the fill layer is highest, while that of the parameter $m$ of the alluvium layer is lowest. Their standard deviation are, respectively, equal to 26% and 1% of the corresponding estimated values. It is because the matric potential data exhibit the lowest sensitivity with $K$ (fill) and the highest sensitivity with $m$ (alluvium) (see Figure 4). Table 1 also shows that the parameters $\alpha$ and $K$

of the fill layer are small, implying that this layer has a strong water-holding capacity, and water will move downward slowly.



Results of the thermal inversion show that the uncertainties of the heat conductivity ($\lambda$) of both fill and alluvium layer are small. This indicates that the heat conductivity parameter is reliably estimated, due to the dense subsurface temperature measurements and the high sensitivity of temperature to the parameter. Table 1 also shows that the heat conductivity of both fill and alluvium layers is relatively

high, which means that the variations of the temperature at the land surface are rapidly propagated downward. The heat conductivity of the alluvium layer is lower than that of the fill layer. This is because a large part of the alluvium layer is saturated with water, and thus has much lower heat conductivity than the drier fill layer.

The coupled inversion results are shown in the last column of Table 1. Compared to the hydrological inversion, the coupled inversion causes the parameter $m$ (fill) to fall by 10%, and $\alpha$ (fill) to rise by 14%. However, it is worth noting that because parameters $\alpha$ and $m$ in the retention curve are negatively proportional, the retention curve does not change much when $m$ decreases and $\alpha$ increases. The heat conductivity of the fill layer exhibits an ignorable change. The table also indicates that while the

uncertainty of the saturation parameter $n$ (alluvium) is smaller (0.5% of the estimated value), the uncertainties of the parameters $n$ (fill) and $\sigma_s$ (fill) are significantly large (26% and 22% of the estimated values, respectively). This can be explained by the fact that the saturation index ($n$) and soil/sediment surface conduction ($\sigma_s$) closely correlate (see Equation 13). As a result, when both of these parameters of the fill layer are concurrently estimated, their uncertainties are higher than in the case of the alluvium

layer, where only the saturation index is estimated (the surface conduction of the alluvium was fixed at $\sigma_s = 0$ S/m).

Comparison of the measured and modeled matric potential of all eight datasets is presented in Figure 5. The figure shows that there is good agreement between measured and modeled data with a Nash-

Sutcliffe efficiency criterion of 0.92. We also observe that the temporal variations of the matric potential over the simulation period mostly occur at the fill layer ($z \leq 1.5$ m). When the depth is equal or greater than 2 m, the matric potential is nearly constant. This suggests that the moisture of the alluvium





layer is less dynamic, and the variations within the fill layer do not frequently propagate to the alluvium layer during the simulation period.

The modeled and measured temperatures at depths from 0.75 to 6 m are shown in Figure 6. The figure indicates the model is capable to reproduces the spatial and temporal variations of the subsurface temperature. The Nash-Sutcliffe efficiency criterion is equal to 0.98. The figure also shows that at the upper depths (0.75 and 1 m), the model slightly underestimates the measurement. This can be explained by the errors of simplification at the land surface boundary. The heat and energy exchanges at the land surface between the atmosphere and land surface were not fully considered. Instead, the land surface temperature was approximated based on the historical data of atmospheric and land surface temperature (see supplement document). The evaporation was represented by the upward flux from the land surface to the atmosphere. The figure also shows that the temporal variation of the measured and modeled temperature data decrease with increasing depth. For example, while the temperature at depth $z =$ 0.75 m varies in a range 8–27°C during the simulation period, it only varies from 11 to 16°C at depth $z = 6$ m. The peaks of the subsurface temperature appear later at deeper locations, as it takes time for heat to flow down.

The measured and modeled apparent resistivity data on May 08, 2013 (when the modeled data were obtained through inversion) are depicted in Figure 7a. The figure indicates that the coupled hydrological-thermal-geophysical simulation effectively reproduces the measured data. Particularly, the lateral variation of the apparent resistivity is simulated with high accuracy. Both measured and modeled data clearly indicate that the upper part of the subsurface section is more conductive (lower resistivity) than the deeper part. This is reasonable, as the deeper section contains more sand and cobbles, while the upper section contains more clayey and silty soils, and therefore is more electrically conductive. Comparison of the measured and modeled resistivity data obtained from the whole simulation period is presented in Figure 7b. The Nash-Sutcliffe efficiency criterion is equal to 0.94. Both Figure 7a and b indicate that the estimation is less accurate for the high resistivity values. This can be explained by the fact that the high resistivity values are located deeper, and thus are harder to fit due to the influence of

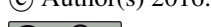



above soil. Another possible reason is that with the same relative measurement error (5%), the measurement error variances of the high resistivity values are larger than those of low resistivity values. As a result, their weights in the objective function (Equation 15) are smaller, and they are less accurately estimated.

The water saturation and temperature versus time over the simulation period at depths $z = 0.025$, 0.475, 0.975 and 1.525 m are shown in Figure 8. For reference, the rainfall data are also presented. The figure shows that the water saturation at the surface layer rapidly responds to variations of the temperature and rainfall. It is relatively wetter at the beginning and the end of the simulation period due to high rainfall

and low temperature (i.e., low evaporation), and drier in the middle due to low rainfall and high temperature. However, the magnitude of the variations of the water saturation quickly decreases with increasing depth. Below z = 0.475 m, the water saturation slowly changes with time, and only gradually increases with a relatively large amount of rainfall in the end of the simulation. Similar to the water saturation, the subsurface temperature exhibits a highly temporal variation in a range 1.8–29.8°C at the

surface but becomes more stable at the deeper depths. At depth $z = 1.525$ m, the temperature varies only from 9.6 to 22.2°C.

The temporal variation of the water flux, which is the sum of the vapor and liquid fluxes versus time over the simulation period at depths from 0.025 to 1.525 m is shown in Figure 9. Comparing Figures Figure 8 and Figure 9, we observe that the temporal variation of the water flux is highly correlated with

that of the water saturation and temperature. The most variation occurs at $z = 0.025$ m, with the flux ranging from –0.001 to 0.024 m/s. At $z = 1.525$ m, the flux is constantly equal to zero. The figure also indicates that the infiltration (positive flux values) is observed at the beginning and the end of the simulation period, when the soil is wet and rainfall occurs. At the middle of the simulation period, when the air temperature is high, the upward flux (negative flux values) occurs because of evaporation. Under

the control of the diffusion, the evaporation can lead to the upward flow up to 1 m depth.

It is worth noting that our study assumed the Rifle subsurface was composed by two homogeneous layers, namely, fill and alluvium. For studies where the spatial heterogeneity is high, we suggest that users construct the model  parameters as spatially-correlated random fields characterized by variogram



functions, and then estimate the parameters (e.g., correlation length, anisotropy value, variance) of these variogram functions as proposed in Finsterle and Kowalsky (2007).

## 4 Effect of temperature dependence of resistivity on hydrogeophysical inversion

In this section, we consider the effects of the temperature dependence of the electrical resistivity on the subsurface hydrological parameters, which are obtained by inverting apparent resistivity data in synthetic isothermal and nonisothermal scenarios. For the isothermal scenario, the temperature was assumed to be constant in time and space at the value averaged over the whole computational domain and over the simulation period. For the nonisothermal scenario, the spatial and temporal variability of the temperature under the influences of the atmospheric temperature and hydrological-thermal parameters was fully considered. It is worth noting that the influences of temperature variability on the electrical resistivity include both direct (temperature-electrical resistivity relationship) and indirect (via changing the hydrological-thermal processes, e.g., gas-liquid phase transition) effects. The synthetic experiment was implemented as below:

1. Run nonisothermal hydrological-thermal-geophysical forward simulation to generate artificial apparent resistivity data. Add Gaussian white noise to the artificial data to obtain the synthetic data.

2. Invert the synthetic apparent resistivity data to estimate the subsurface hydrological parameters, assuming that the subsurface temperature is spatiotemporally constant (isothermal scenario).

3. Invert the synthetic apparent resistivity data to estimate the subsurface hydrological parameters considering the nonisothermal process (nonisothermal scenario).

4. Compare inversion results of the two scenarios to evaluate the effect of the subsurface temperature variability on the hydrogeophysical inversion.

The computational domain, model parameters, and initial and boundary conditions for the synthetic forward simulation were taken from the coupled hydrological-thermal-geophysical inversion as presented in Section 3. Because the variation of the water saturation mostly occurs in the fill layer, we focused on estimating the hydrological parameters of this layer, including $\alpha$, $m$ and absolute permeability $K$. For both isothermal and non-isothermal scenarios, the initial guesses for the three





parameters were set at $\alpha = 1.8 \times 10^{-5}$ (Pa$^{-1}$), $m = 0.4$ and $K = 4.5 \times 10^{-15}$ m$^2$. To increase the sensitivity of the apparent resistivity data with the hydrological parameters, we selected four synthetic apparent resistivity datasets corresponding with high water saturation values. The Gaussian noise, with a mean of zero and a relative standard deviation of 5%, was added to the artificial apparent resistivity data to

generate synthetic data for the hydrogeophysical inversion.

Comparison of the synthetic van Genuchten water retention curve and the ones obtained by the isothermal and nonisothermal hydrogeophysical inversion is exhibited in Figure 10a. Although the nonisothermal hydrogeophysical inversion does not perfectly estimate the synthetic parameters (due to

the nonuniqueness and the correlation between parameters), its estimation is close to the synthetic ones. Meanwhile there is a large difference between the synthetic and estimated curves obtained by the isothermal hydrogeophysical scenario.

Figure 10b presents the synthetic and modeled apparent resistivity data using a 1:1 plot. The figure

shows that the nonisothermal scenario better reproduces the synthetic apparent resistivity than the isothermal does. Correlation, bias, and RMSE between the synthetic and nonisothermal-simulated electrical resistivity data are 0.98, 1 and 2.29, respectively, while these criteria for the isothermal scenario are 0.96, 0.98 and 3.54. In brief, ignoring temperature variability and its influence on electrical resistivity in the hydrogeophysical inversion is very likely to cause a large error for the model parameter

estimation, and to reduce agreement between modeled and measured geophysical data.

## 5 Summary and Discussion

We developed a coupled hydrological-thermal-geophysical inversion scheme that quantifies the dependence of the electrical resistivity on both subsurface moisture and temperature, instead of solely moisture, as has been typical for previous hydrogeophysical inversion schemes. This scheme permits

simulation of nonisothermal, multiphase subsurface heat and water fluxes, as well as the relationship between temperature, moisture and electrical resistivity. It accounts for the spatiotemporal variability of moisture and temperature in the shallow subsurface, and can include multiple geophysical and non-





geophysical measurement constraints. At present, TOUGH2 cannot simulate the land surface processes and energy balance at the land surface. To mitigate this disadvantage, this study approximated the top land surface temperature boundary condition from the atmospheric temperature using a regression approach. The evaporation was considered via the gas phase of moisture. The evaporation rate was
simulated as the water vapor fluxes moving upward from the top layer to the atmosphere.

The new approach was applied to data collected at a field site in Rifle, Colorado. The ERT data were used to characterize subsurface stratigraphy, and to constrain the computational domain for the hydrological-thermal model. The time-lapse ERT data were used with other hydrological and thermal
data to constrain the inversion. The inversion results show that our developed scheme well reproduces the matric potential, temperature, and apparent resistivity data. The obtained results indicate that the temporal variation of the moisture mostly occurs at the overlying fill layer, due to the relatively small amount of rainfall and the high water-holding capacity of this layer. The alluvium moisture exhibits a minimal change. Both fill and alluvium layers have high thermal diffusivities, permitting the variation
of the air temperature to rapidly move down. The obtained results also indicate that the heat conductivity and van Genuchten parameters of both fill and alluvium layers are well estimated with low uncertainties. However, due to limited temporal variations of moisture content (and thus ERT data), it is difficult to obtain the absolute permeability of the fill layer and the petrophysical parameters.

To evaluate the influence of the temperature dependence of the electrical resistivity on the estimation of the hydrological parameters in the hydrogeophysical inversion, we performed a synthetic study. By comparing the results obtained from the isothermal and nonisothermal scenarios, we determined that ignoring the spatial and temporal variability of the subsurface temperature may cause errors in the estimation of hydrological parameters.

Our study documents the value of accounting for the dependence of both moisture content and temperature on electrical resistivity within a hydrological-thermal-geophysical inversion framework. The inversion scheme presented here can be widely applied to many studies  striving to quantify





hydrological and thermal dynamics in the subsurface. We believe that this and other approaches that permit rapid assimilation of autonomous monitoring datasets will greatly improve our understanding of terrestrial system properties and their behaviour, including their response to environmental perturbations such as floods and droughts.

**Acknowledgments**. This material is based upon work supported as part of the Sub-surface Science Scientific Focus Area funded by the U.S. Department of Energy, Office of Science, Office of Biological and Environmental Research under Award Number DE-AC02-05CH11231. The authors would like to thank Stefan Finsterle for providing iTOUGH2 codes and support, and Thomas Gunther for providing
the BERT codes.

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





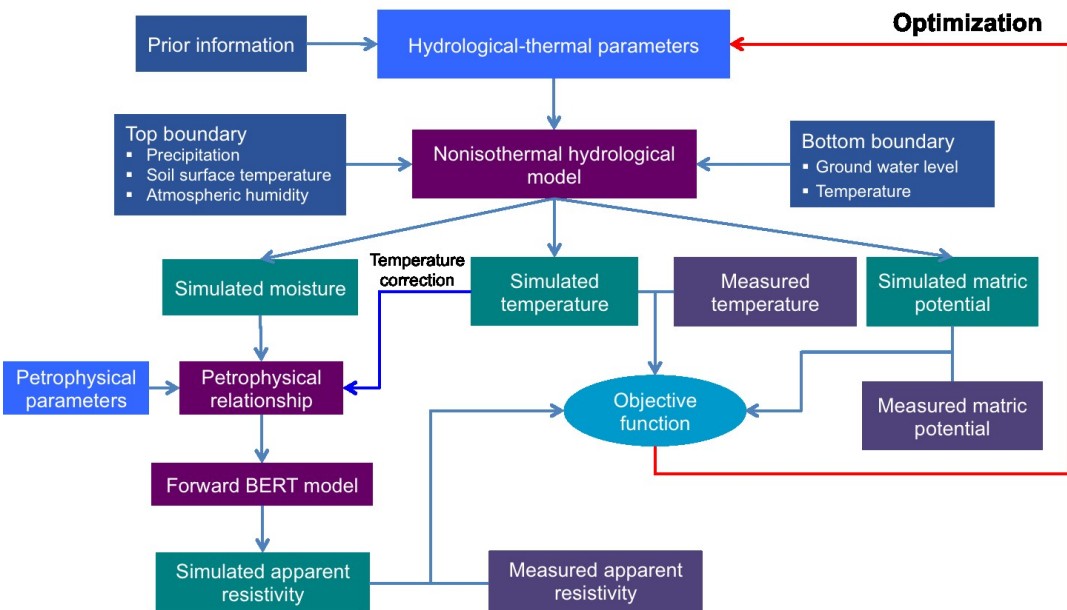

**Figure 1:** Flowchart showing the steps involved in the coupled hydrological-thermal-geophysical inversion scheme. The objective function is represented by Equation (15). Estimated parameters include hydrological (permeability, van Genuchten curve parameters), thermal (heat conductivity) and petrophysical (Archie' model) parameters. The navy blue rectangles denote the model inputs, including prior information about estimated parameters, and the top and bottom boundary conditions. The purple rectangles denote the forward hydro-thermal, geophysical and petrophysical models. The teal and indigo rectangles, respectively, denote the simulated and measured data for the parameter estimation.




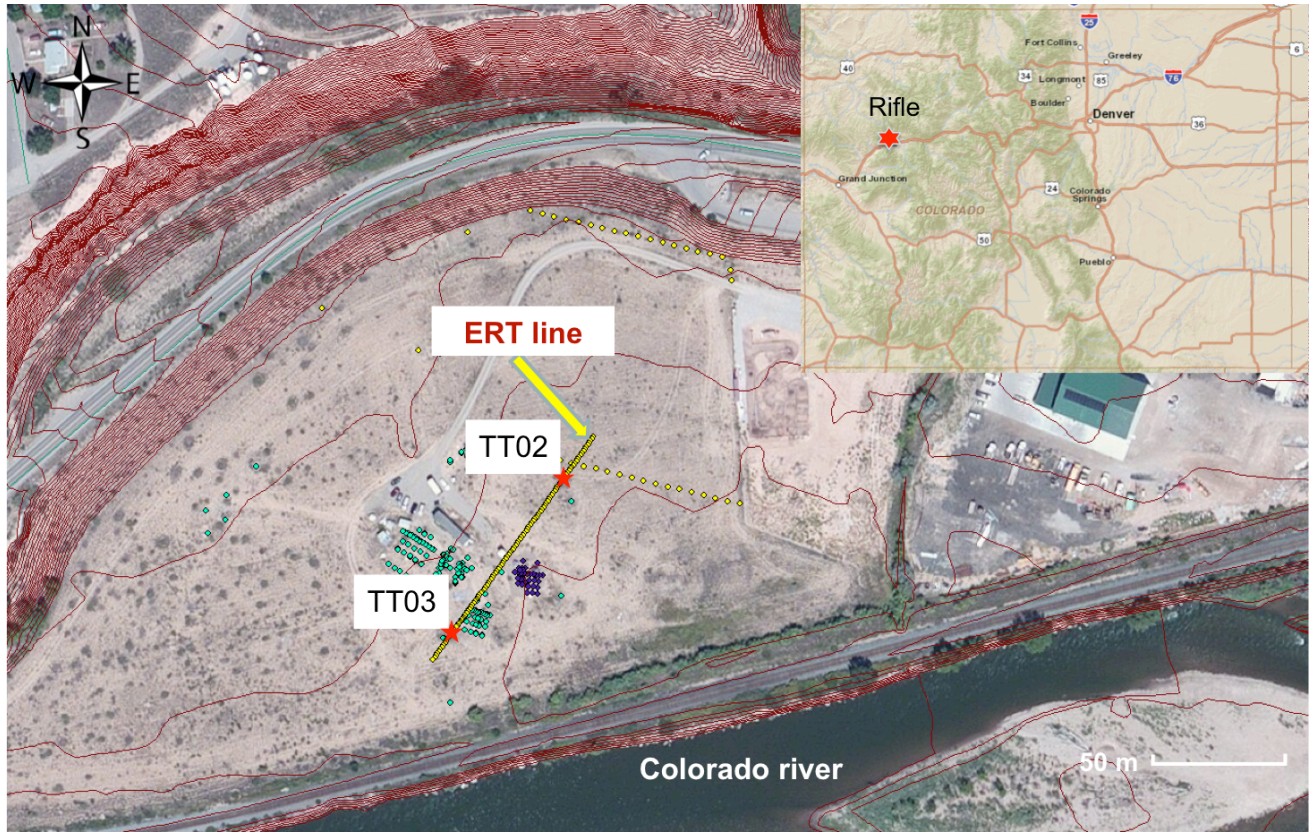

**Figure 2:** Plan view of the Rifle floodplain of the Colorado River, Colorado, and the location of the TT02 and TT03 wells and ERT line. The top right figure shows the location of the study site in Colorado.





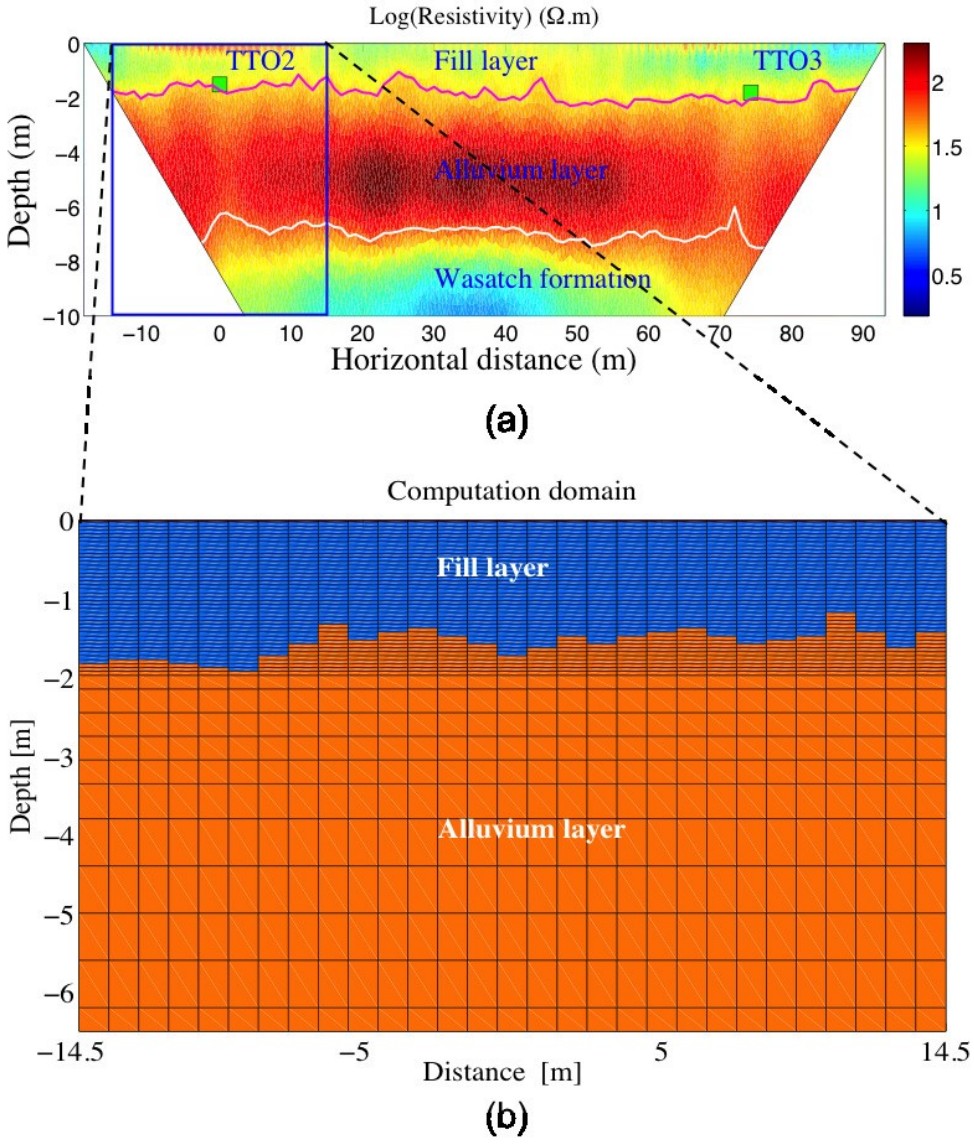

**Figure 3:** (a) The 2-D image of the soil electrical resistivity obtained by inverting ERT data collected on May 20, 2013. The magenta and white lines delineate the fill-alluvium and alluvium-Wasatch boundaries. Green square markers denote the fill-alluvium boundary determined from the well logs of TT02 and TT03 and adjacent wells, as recorded in the field during drilling. The blue rectangular box indicates the hydrological-thermal computational domain. (b) Computational domain for the hydrological-thermal inversion with associated grid mesh. Blue and orange regions represent the fill and alluvium layers, respectively. The domain is situated below an atmospheric layer (top boundary) and above the relatively impermeable Wasatch (bottom boundary).





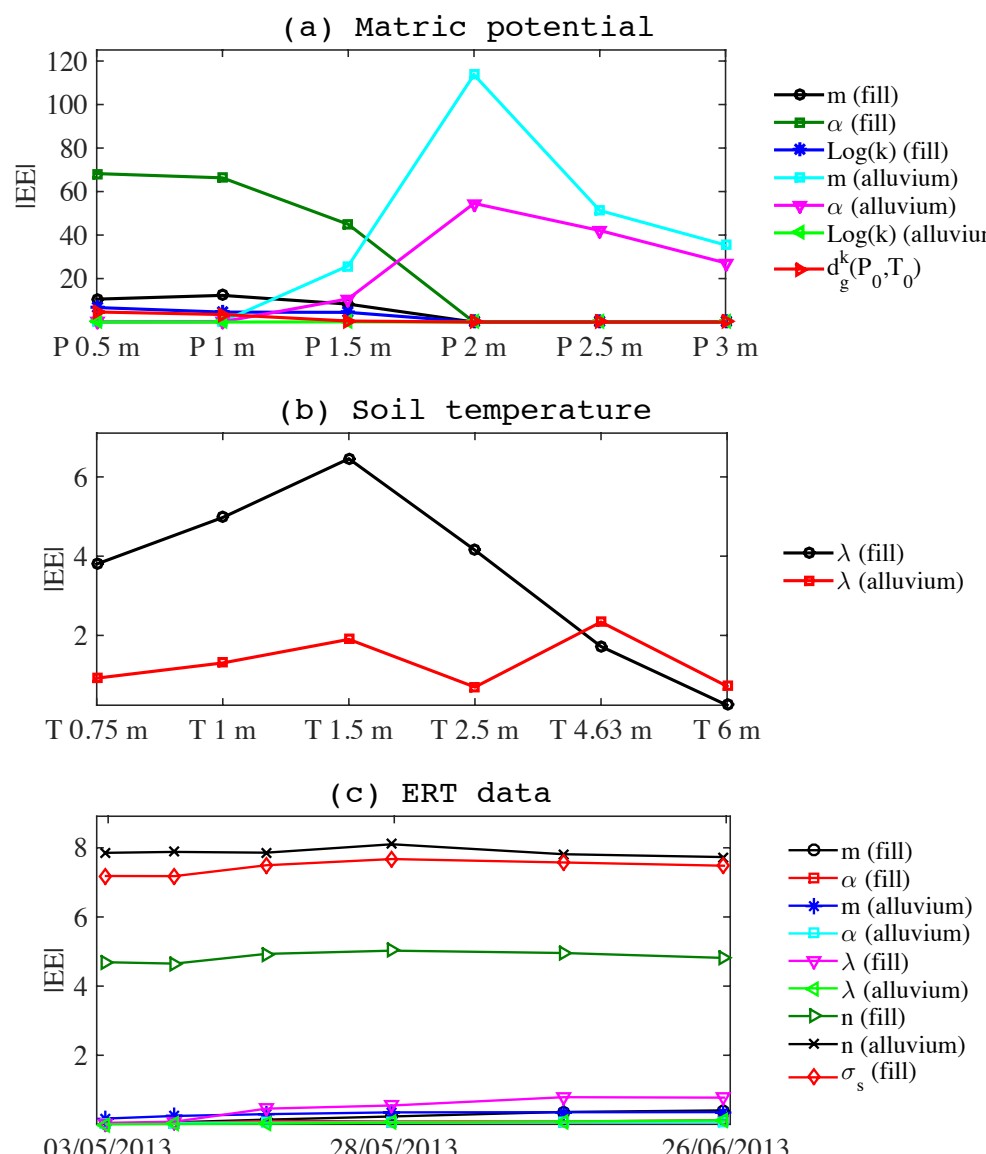

**Figure 4:** (a) The sensitivity coefficient |*EE*| of the matric potential at depths of 0.5, 1, 1.5, 2, 2.5 and 3 m, with respect to the hydrological parameters of the fill and alluvium layers, and the gas diffusion coefficient standard conditions. (b) The |*EE*| of the temperature at depths of 0.75, 1, 1.5, 2.5, 4.63 and 6 m, with respect to the heat conductivity of fill and alluvium layers. (c) The temporal variations of the |*EE*| of the resistivity data with respect to the soil hydrological-thermal and petrophysical parameters of both fill and alluvium layers.





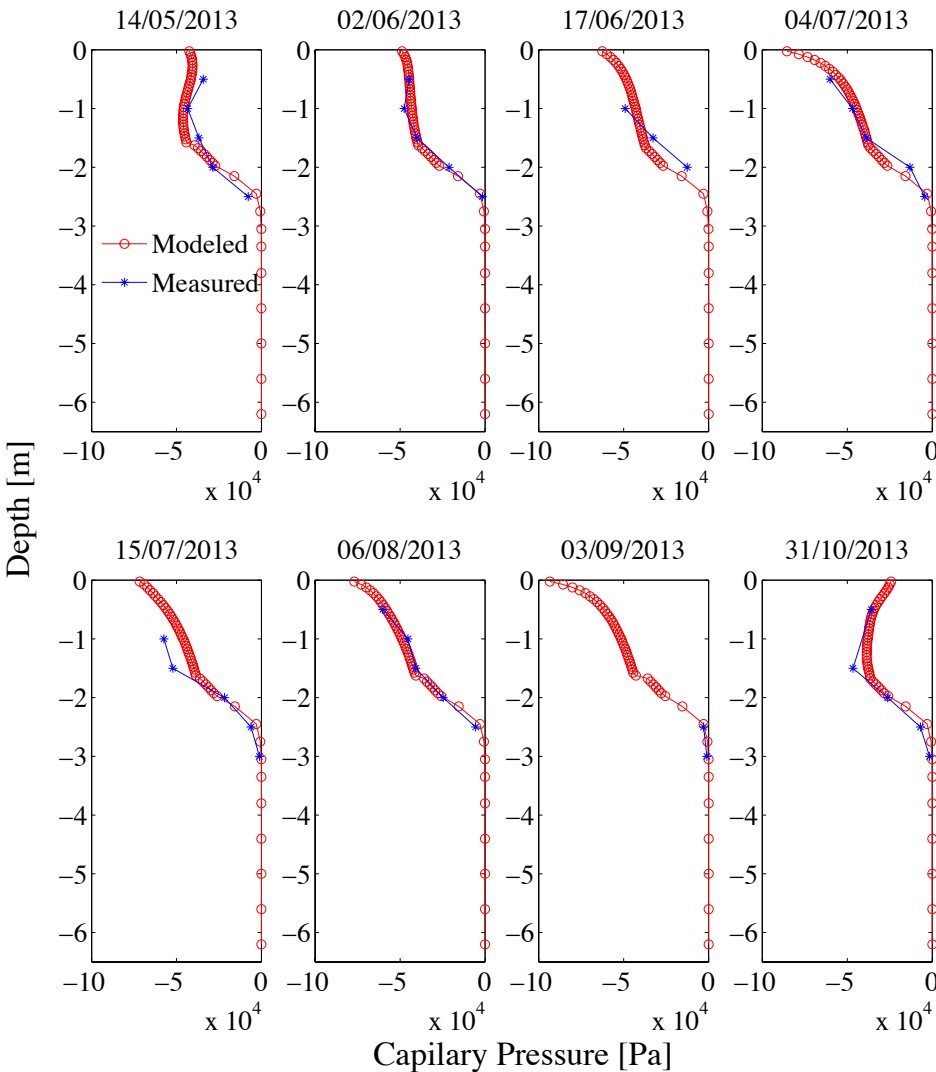

**Figure 5:** Comparison of the measured and modeled matric potential data for all measurement occasions. The red symbols represents the modeling results obtained from the coupled hydrological-thermal-geophysical inversion. The blue symbols denote the measurements.





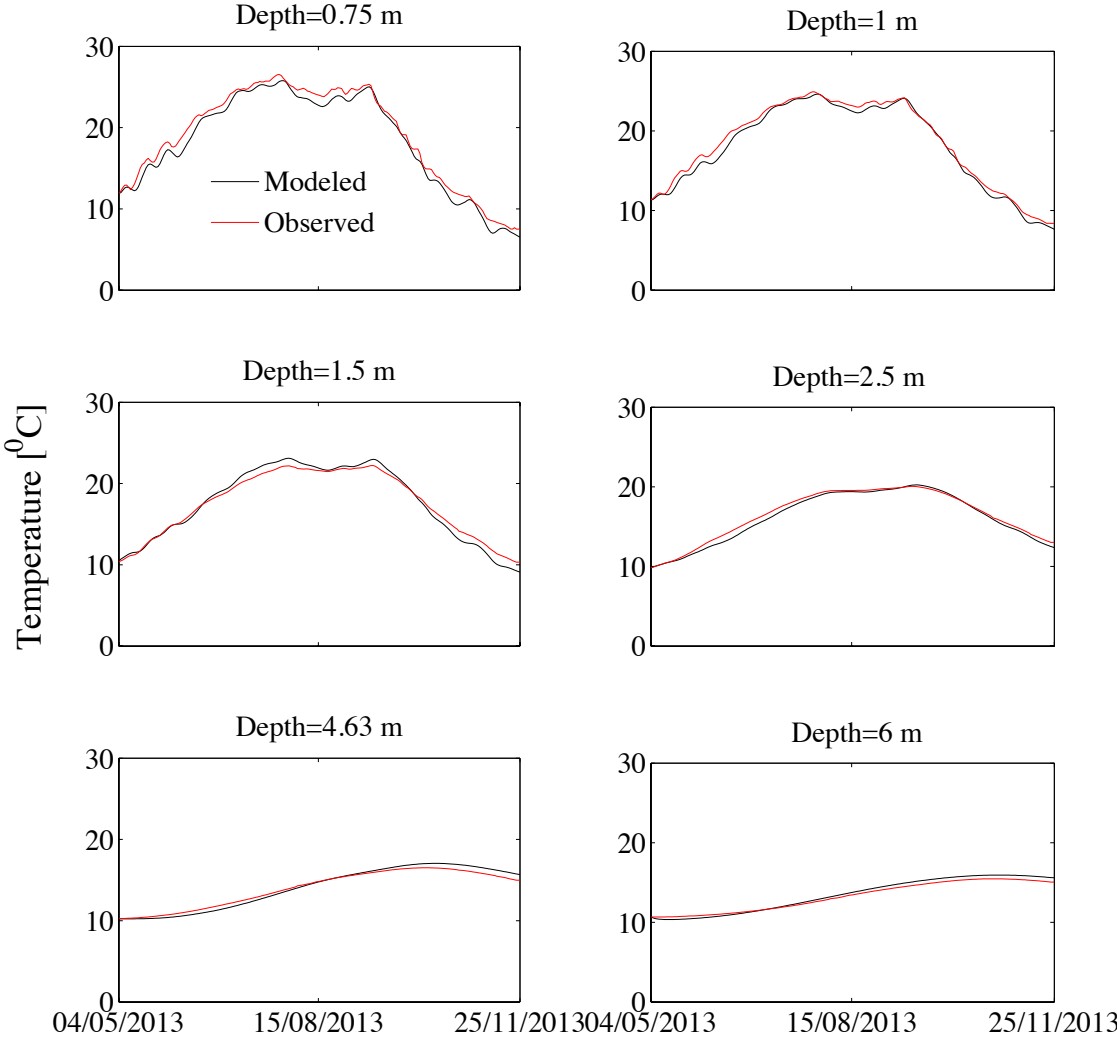

**Figure 6:** Comparison of the measured and modeled temperatures at depths of 0.75, 1, 1.5, 2.5, 4.63 and 6 m during the simulation period. The black line denotes the modeling results obtained from the coupled hydrological-thermal-geophysical inversion. The red line represents the measurements.




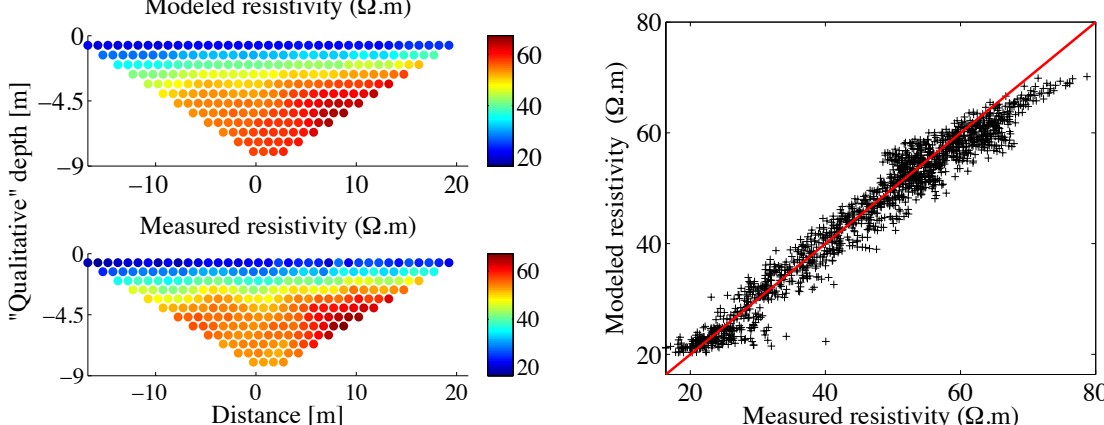

**Figure 7:** (Left) An example of "quantitative" plots of the modeled and measured apparent resistivity data on 08/05/2013. (Right) Comparison of all measured and modeled apparent resistivity data in a 1:1 plot. The modeled data were obtained from the coupled hydrological-thermal-geophysical inversion.





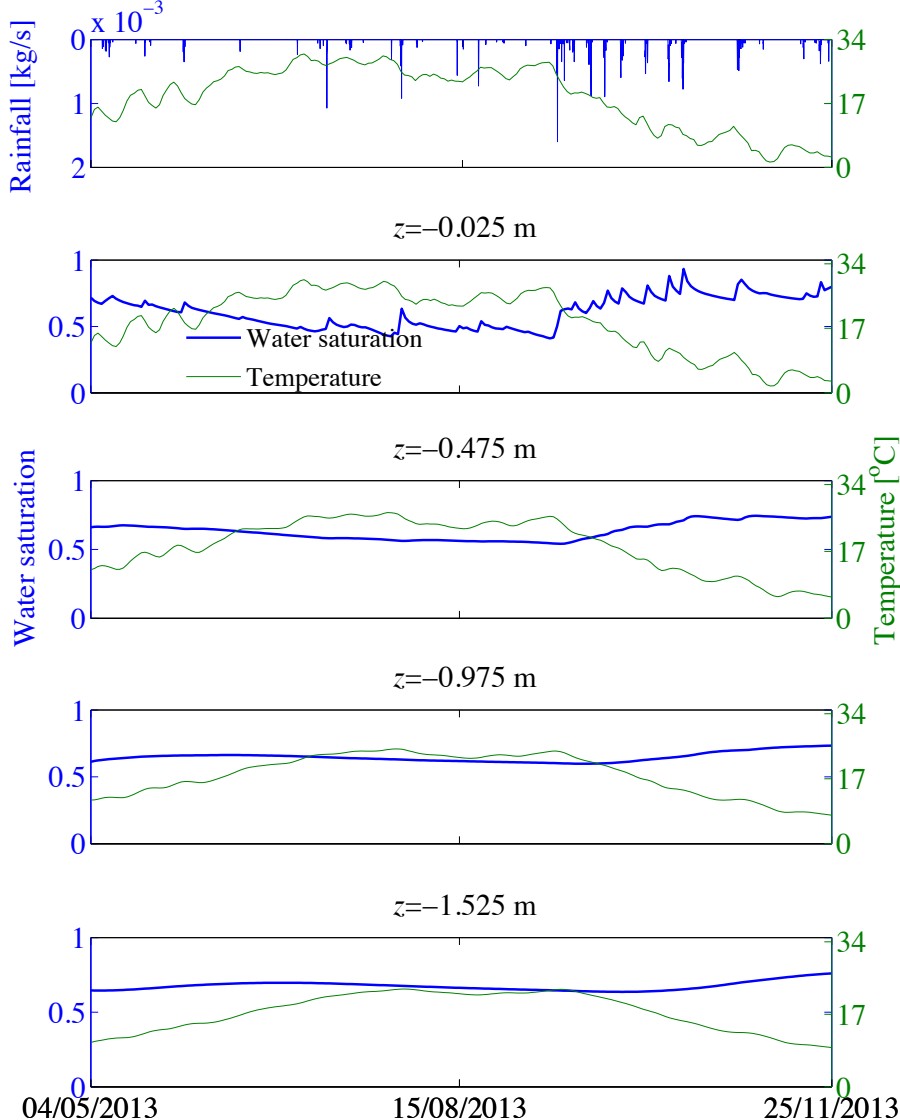

**Figure 8:** Temporal variation of the simulated water saturation and temperature at depths z =0.025, 0.475, 0.975 and 1.525 m of the TT02 well (center of the computational domain). For reference, rainfall and soil surface temperature data are also plotted.





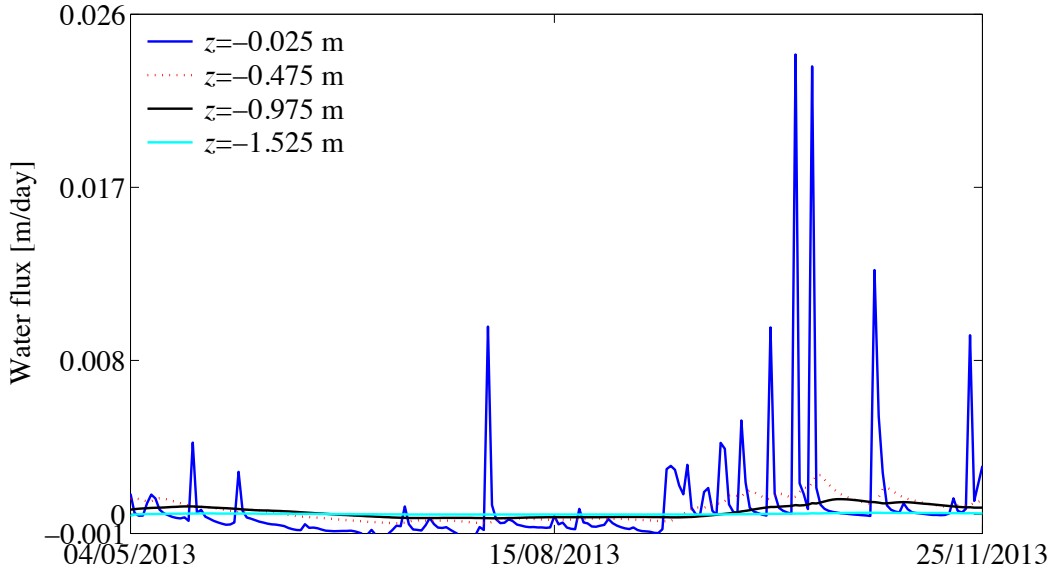

**Figure 9:** Temporal variation of the simulated water flux at depths z =0.025, 0.475, 0.975 and 1.525 m of the TT02 well. The positive and negative values indicate the downward and upward flows.

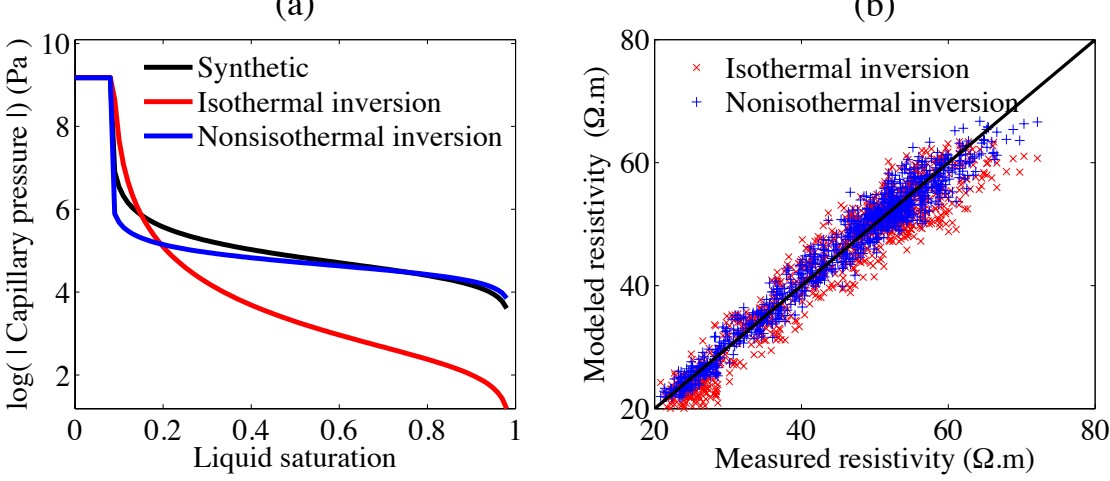

5   **Figure 10: (**a) Comparison of the synthetic and estimated van Genuchten's retention curve. (b) Comparison of synthetic and modeled apparent resistivity data. The red color represents the results obtained by the isothermal hydrogeophysical inversion scenarios.The blue color denotes the nonisothermal scenario.





**Table 1: Constraints and estimated values of the hydrological-thermal and petrophysical parameters for different inversion cases.**

| Parameter | Range | Hydrological inversion (HI) | Thermal inversion (TI) | Coupled inversion |
|---|---|---|---|---|
| $m$ (fill) (-) | $0.4 - 0.6$ | $0.516$ ($\pm0.039$, 7.5%) | From HI | $0.464$ ($\pm0.005$, 1.1%) |
| $\alpha$ (fill) (Pa$^{-1}$) | $10^{-5} - 10^{-4}$ | $2.680\times10^{-5}$ ($\pm2.030\times10^{-6}$, 7.6%) | From HI | $3.054\times10^{-5}$ ($\pm3.951\times10^{-7}$, 1.3%) |
| $K$ (fill) (m$^2$) | $10^{-15} - 10^{-13}$ | $8.684\times10^{-15}$($\pm2.235\times10^{-15}$, 25.7%) | From HI | From HI |
| $m$ (alluvium) (-) | $0.1 - 0.3$ | $0.184$ ($\pm0.002$, 1.2%) | From HI | From HI |
| $\alpha$ (alluvium) (Pa$^{-1}$) | $10^{-4} - 10^{-1}$ | $0.039$ ($\pm0.003$, 8.3%) | From HI | From HI |
| $D(P_0, T_0)$ (m$^2$s$^{-1}$) | $10^{-5} - 10^{-7}$ | $7.070\times10^{-5}$ ($\pm2.893\times10^{-6}$, 4.1%) | From HI | From HI |
| $\lambda$ (fill) (Wm$^{-1}$°C$^{-1}$) | $1.2 - 2.7$ | - | $2.409$ ($\pm0.016$, 0.7%) | $2.447$ ($\pm0.015$, 0.6%) |
| $\lambda$ (alluvium) (Wm$^{-1}$°C$^{-1}$) | $1.2 - 2.7$ | - | $1.423$ ($\pm0.029$, 2%) | From TI |
| $n$ (fill) (-) | $1.3 - 2.5$ | - | - | $2.222$ ($\pm0.574$, 25.8%) |
| $n$ (alluvium) (-) | $1.3 - 2.5$ | - | - | $1.437$ ($\pm0.007$, 0.5%) |
| $\sigma_s$ (fill) (S/m) | $0.02 - 0.05$ | - | - | $0.043$ ($\pm0.010$, 22.2%) |

- $m$ and $\alpha$ represent the pore size distribution of the soil and reciprocal of the air-entry pressure (Equation 7)
- $K$ is the absolute permeability (Equation 5)
5
- $D(P_0, T_0)$ is the is the gas diffusion coefficient at the standard condition ($P_0$ = 1 atm and
- $T_0$ = 0°C) (Equation 6)
- $\lambda$ is the heat conductivity (Equation 12)
- $n$ and $\sigma_s$ are the saturation index and soil surface conduction (Equation 13)

