# Peer review of "Quantifying Shallow Subsurface Water and Heat Dynamics using Coupled Hydrological-Thermal-Geophysical Inversion"

_Hydrology and Earth System Sciences, 2016_

## Referee Comment (RC1) · N. Linde (Referee) · 4 May 2016

The manuscript of Tran et al. address important questions that needs to be addressed for efficient and reliable integrated terrestrial Earth monitoring and interpretations using geophysical data. They couple a state-of-the-art two-phase (gas and liquid) and three-component (air, water, heat) simulator and its associated inverse capabilities with an electrical resistivity tomography (ERT) forward solver. By inferring the values of the petrophysical parameters they can use the ERT data together with point-measurements of state variables to infer various properties of the subsurface. A synthetic test example shows that ignoring the influence of temperature on the resulting electrical conductivity leads to biased estimates of hydrodynamic properties. They also

present an application to the Rifle site, Colorado, and show that the method is capable of parameterizing a subsurface model such that it can make quantitative predictions of various state variables. What I like with this paper is that it highlights both the need and the challenges of fully-coupled hydrogeophysical inversion. Clearly, multi-phase and several components must be included. This leads to many unknowns to estimate and the estimation is quite difficult to achieve due to non-linearity and the limited information content in the available data. Also, I do appreciate the attempts of using a global sensitivity analysis to enable the parameter estimation to focus on the most important parameters. I judge that this would be a very nice contribution to HESS after moderate revisions.

Comments: 1. It is a bit surprising that variations in solute concentrations are not considered. One would expect that the rainwater is much more resistive than older formation water. I realize that this is challenging, but mentioning this in the discussion as an outlook might be appropriate. Also, there is no information (at least I did not see it) about what salinity/electrical conductivity of the pore water that was assumed. I assume that a constant value was chosen. Was it based on actual water samples or was it a fitting parameter to get the right amplitudes in the resistivity model? This needs to be clarified.

2. The hydrological model contains simply two geological layers with uniform properties. Clearly, the ERT response must consider the deeper (aquitard) Wasatch formation. I assume that the authors do this, but again, this is not stated in the manuscript. In short, the model domain is different for the electrical problem and this must be clarified. Perhaps the authors embed the simulated resistivities in the fill and alluvial layer into the model shown in Figure 3a?

3. The only lateral heterogeneity in the model is due to a spatially varying interface between the alluvial fill and the alluvial layer. Clearly, any ERT delineation of this interface will be highly uncertain (especially as the water table divides the gravelly sand into a saturated and an unsaturated part. I assume that the interface was guided by an

observed interface in well TT02? This should be explained and the sensitivity to errors in this interface should be discussed in the paper.

Minor comments: Page 1, Lines 18-19: Are the measured data relatively accurate and the simulated data reproduce them (what is written) or is it the predictions that are relatively good (what I assume the authors want to write). Please correct. Also, try to be quantitative and replace "relatively" with quantitative measures.

Page 2, line 4: Perhaps "play an important role in controlling" instead of "control". Other factors control this partitioning as well.

Page 2, line 5: Is ecosystem moisture an accepted term?

Page 2, line 10: Perhaps "spatio-temporal". Reconsider the term "native". Generally speaking, this sentence would be easier to follow by clarifying what are the differences in scales between the so-called native processes and ecosystem functioning.

Page 2, line 12: I suggest to remove "direct measurements". Soil moisture is typically inferred from dielectric properties. This is not a direct measurement. Indeed, one can discuss if direct measurements exists altogether.

Page 2, lines 21-24: Data and models are not the same! Hubbard et al. (2001) integrate geophysical models/tomograms with point measurements. On line 24, add "pumping tests" as these are needed to get the average hydraulic conductivity, while the flowmeter gives relative variations.

Page 3, line 2: I think it should be "hillslope".

Page 3, lines 3-4: Unfortunately, few hydrogeophysical inversion approaches are developed to "improve quantification of subsurface processes". They are in most case only used for parameter estimation, which is something very different and much more restrictive.

Page 3, line 6: You need a qualifier before inversion, something like "hydrological".

Page 3, line 13: I don't think that "its" is used correctly here. Please rephrase.

Page 3, line 24: "at high resolution (add references)".

Page 4, line 1: This is true for near-surface applications (please clarify). Seismological stations are probably used much more for geophysics in general. I would call ERT a method, not an approach.

Page 4, line 12: Replace "and" with "which". Line 17, remove "the", line 18 change to "images". Line 26, replace "into a" with "within".

Page 5, line 7: I would remove references to Archie here. Archie defined the formation factor and the effect of saturation. However, referring to Archie implies that surface conductivity is ignored, which is not done here.

Page 6, line 8: Change symbol for porosity to be consistent with the rest of the paper.

Page 6 and elsewhere: Equations are part of the sentences, so use "," and "." as appropriate after the equations. Ensure that variables are in italics. At the moment, they are sometimes, but not always (e.g., line 19).

Page 7, line 18: Perhaps extend this aspect in the discussion? Is this a big issue?

Page 8, line 11: Why not use subscript "b", as sigma_a or rho_a suggest apparent properties and not bulk properties.

Page 8, lines 17-18: It is ok to fix the cementation index, but I am not sure if I see the argument here. The surface contribution is a constant over time while the saturation contribution is time-variable, so identifiability should not be an issue if enough variations in the states are probed.

Page 9, lines 24-29: How did you balance the different types of data? Ideally, one should have a WRMSE of 1 for each data type after inversion and using actual data errors. In practice, this is essentially never done. The ERT data are given errors of 5% to somehow accommodate modeling/petrophysical errors, but they are presented as

observational errors while the actual observational errors are probably less than 1%. I assume similar common "tricks" are used for the other data types. I don't criticize this, but some discussion about how the data were weighted and reporting the final data misfits for the individual data types should be part of the paper.

Page 12, line 3: "at the within". Please revise.

Page 13, line 13: It should be Günther. Also Rücker when his name is used.

Pager 13, lines 15-18. This is a bit of a circular argument (a similar construct is also found later in the paper). Why not write something like: As expected, the clay-rich fill and Wasatch layers show up as less resistive layers in the ERT . . .".

Page 13, line 18: Why not across the full ERT line?

Page 13, lines 21-22. Give references that support these values.

Page 13, line 23: This is ok for the hydrological model, but clearly insufficient for the ERT modeling. Express how you extend the ERT model and account for the Wasatch layer.

Page 14, line 2: "the meteorological".

Page 14, line 13: Write "apparent resistivity".

Page 14, line 14: State what this implies in terms of assumptions. Using data from a well that is not located in the modeled study area.

Everywhere: Please consider to replace "heat conductivity" with the more common term: "thermal conductivity".

Page 18, line 14: Replace "ignorable" with "negligible".

Page 19, line 5: It should be "reproduce".

Page 19, line 27: Write "high apparent resistivity values", same for line 28. Also, on line 28: replace "located deeper" with "more sensitive to deeper locations".

Page 20, lines 1-5: This problem would not appear in a log-log plot and it would be more consistent with the error model.

Page 20, line 21: It seems impossible that the downward flux is 24 mm/s. Please clarify what this value refers to? I interpret this in the same way as a Darcy flux and this seems unrealistically high.

Page 20, line 25: Replace "to the upward flow up to" with "upward flow starting at".

Page 21, line 5: Add "estimated" before "subsurface".

Page 21, line 15: State how much noise was added.

Page 22, lines 22-24: This is a worthy contribution, but it seems as if not accounting for the variations of salinity is a bit limiting. The precipitation will clearly be much more resistive than the groundwater. Perhaps something to comment on?

Page 24, lines 1-4: Perhaps comment on the potential of filtering approaches/data assimilation in the context of this type of problems?

Figure 1: What are the East-West trending yellow dots. What are the green dots. Explain or remove.

Figure 2: The depth is positive with depth, so please remove "-" signs". Also give units and properties shown (log10 of resistivity). Add "inferred" before "fill-alluvium". Also, explain how this was done somewhere in the paper.

Best wishes,

Niklas Linde
* * *

---

## Referee Comment (RC2) · Dr Boaga (Referee) · 17 May 2016

General Comments The paper concerns an interesting and innovative procedure of coupled inversion for hydrological and geophysical parameters. In particular Authors present double phases air-water-heat inversion coupled with ERT data. Tran et al. show the results for both synthetic test and real data, with a relevant sensitivity analysis of the parameter estimation which highlights state variables resolution capabilities. The topic is of interest for HESS readers and the manuscript is concise and well written. Despite this, I suggest minor revision before the acceptance for publication: some points need to be better clarified, some figures should be revised cause they are not at the level of the paper, some captions need improvement. Here below a list of detailed

[Figure]

comments.

Specific Comments The Manuscript comes without a continuing lines numbering, restarting in every pages. This usually does not help the revision work. I will refer to page number and relative line numbering. Pg3 Ln 20-25 Sentences not clear to me. You cannot assert that difficulties in hydrogeophysics are linked to temporal/spatial resolution variables problems, and then refer to the high resolution of autonomous acquisitions (?). As in the paper of Binley et al. you cited, several methods are applied second their proper potentials. On the other hand I agree with you we are still far in properties characterization, and your paper represents a relevant step forward. Pg4 Ln 8. I suggest suspension points after properties affecting resistivity description, cause it is not (and cannot be) exhaustive. Ln 24. I'm always prudent when reading there are no similar attempts in the previous literature. Some different works partially fronted the topic (e.g. Jardani et al 2013; Irving et . 2010), but this does not affect the quality of yours work. Probably the differences stay in the design of coupled hydro-geophysical inversion, and the level of results obtained. Pg6 There is probably an error in the symbol of porosity in Eq2 or in the text. Pg9 ln3. There's no need to introduce the relation between resistivity and conductivity here.Ln 10-15. (fig1). The flowchart is unclear and the caption confusing. I understand it is ambitious describe graphically all the inversion scheme, but I suggest to redrawn the flowchart and explain better this crucial part of your relevant work. Pg10.Ln20. I do not understand the sentence about initial guesses, it sounds unnecessary. Explain simply your (elegant) procedure. Pg12Ln5 (fig2). Map figure should be re-drawn, it is not well readable, missing coordinates, label and fonts are too small Pg13.Ln1 What is the relationship with yours work and Arora et al. one? Not clear why do you present extensively this biogeochemical topic here. Ln 18. Why the hydrological model contains simply two geological layers? This is not specified. Fig.3 should be revised. Please put unit over the scale and not with X label. It is unusual the starting with negative distance values. Ln 19 Please insert citation introducing Wasatch layers resistivity, otherwise these seems reasonable but not supported considerations (and should be placed in the discussion section). Ln 23.

[Figure]

Please introduce before the characteristics (e.g. a stratigraphy description) of TT02 and TT03 boreholes. Pg14 Ln 11. How do you approximate bottom temperature from land surface temperature? Pg15 Ln13. Some confusion between m (meter) and matric potential symbols. Fig.4 Caption should be improved to explain better these relevant sensitivity graphs. Tab.1 Table is quite confusing to me and caption does not help. Explain better what's from hydrological inversion and what's from the coupled one. Note: 'To' miss the initial bracket?

Thanks for the reading of a very interesting and well written work, which in my opinion opens important further perspectives for hydrological characterization.
* * *

---

## Author Comment (AC1) · 20 Jul 2016

We are grateful Dr. Niklas Linde for his careful comments, corrections and suggestions. His valuable review helped us much to improve our manuscript.

The manuscript of Tran et al. address important questions that needs to be addressed for efficient and reliable integrated terrestrial Earth monitoring and interpretations using geophysical data. They couple a state-of-the-art two-phase (gas and liquid) and three-component (air, water, heat) simulator and its associated inverse capabilities with an electrical resistivity tomography (ERT) forward solver. By inferring the values of the petrophysical parameters they can use the ERT data together with point measurements of state variables to infer various properties of the subsurface. A synthetic test example shows that ignoring the influence of temperature on the resulting electrical conductivity leads to biased estimates of hydrodynamic properties. They also present an application to the Rifle site, Colorado, and show that the method is capable of parameterizing a subsurface model such that it can make quantitative predictions of various state variables. What I like with this paper is that it highlights both the need and the challenges of fully-coupled hydrogeophysical inversion. Clearly, multi-phase and several components must be included. This leads to many unknowns to estimate and the estimation is quite difficult to achieve due to non-linearity and the limited information content in the available data. Also, I do appreciate the attempts of using a global sensitivity analysis to enable the parameter estimation to focus on the most important parameters. I judge that this would be a very nice contribution to HESS after moderate revisions.

Comments: 1. It is a bit surprising that variations in solute concentrations are not considered. One would expect that the rainwater is much more resistive than older formation water. I realize that this is challenging, but mentioning this in the discussion as an outlook might be appropriate. Also, there is no information (at least I did not see it) about what salinity/electrical conductivity of the pore water that was assumed. I assume that a constant value was chosen. Was it based on actual water samples or was it a fitting parameter to get the right amplitudes in the resistivity model? This needs to be clarified.

Reply: In our case study in Rifle, the electrical conductivity of pore water does not much change in time and space. The measured water electrical conductivity ranges from 0.237 to 0.255 S/m at two different wells over the simulation period. Therefore, we selected the average value of these samples σ = 0.244 S/m. We presented this information in line 12-13, page 8 as below:

*"The electrical conductivity of water, which does not vary significantly over time at the Rifle site, was taken from the measurements at the nearby wells, and is equal to 0.244 S/m."*

We also discussed the approach that considers the solute dynamics in hydrogeophysical inversion in the revised version (lines 13-16 page 8) as below:

*"In case that the spatio-temporal variation of solute concentration and resulting electrical conductivity in pore water are significant, it dynamics should be simulated by considering it as a component in TOUGH2. A formula that links the solute concentration*

*with the water electrical conductivity is also needed to develop."*

2. The hydrological model contains simply two geological layers with uniform properties. Clearly, the ERT response must consider the deeper (aquitard) Wasatch formation. I assume that the authors do this, but again, this is not stated in the manuscript. In short, the model domain is different for the electrical problem and this must be clarified. Perhaps the authors embed the simulated resistivity in the fill and alluvial layer into the model shown in Figure 3a?

Reply: We discuss how to map the apparent resistivity from hydrological to geophysical computational mesh in the revised version (lines 5-12 page 14) as below:

*"The BERT computational mesh was automatically generated in BERT. We set the maximum cell size, which controls the mesh refinement at a small value (0.2 m) to capture the local variation of the soil electrical resistivity. Other parameters that determine the BERT computational mesh were kept as their default values. For more information about BERT mesh generation, we refer to Rücker et al. (2006). The apparent resistivity was mapped from the hydrological to the BERT mesh using the nearest distance method, i.e., a cell in the BERT mesh will get the resistivity value of its nearest cell in the hydrological mesh. The electrical resistivity of the Wasatch layer was set at its average value obtained from geophysical inversion ($\rho_b^{Wasatch} = 45\ \Omega.m$)."*

3. The only lateral heterogeneity in the model is due to a spatially varying interface between the alluvial fill and the alluvial layer. Clearly, any ERT delineation of this interface will be highly uncertain (especially as the water table divides the gravelly sand into a saturated and an unsaturated part. I assume that the interface was guided by an observed interface in well TT02? This should be explained and the sensitivity to errors in this interface should be discussed in the paper.

Reply: Yes, we selected the threshold of electrical resistivity for determining the interface between layers using observation at TTO2 and TTO3 wells as references. We added this information in the revised version (lines 15-22 page 13) as below:

*To specify the locations of the fill-alluvium and alluvium-Wasatch interfaces from ERT geophysical inversion, we used the depths of these interfaces observed at the TTO2 and TTO3 well as the references to determine resistivity thresholds. Accordingly, a grid cell with a resistivity greater than 1.52 $Log_{10}$ ($\Omega.m$) and above 1.5 m depth belongs to the alluvium layer. The cells whose resistivity values are smaller than 1.83 $Log_{10}$ ($\Omega.m$) and below 5 m are assigned to the Wasatch layer. The remaining cells are in the fill layers. The magenta and white lines in Figure 3 represent the fill-alluvium and alluvium-Wasatch interfaces, respectively.*

Minor comments: Page 1, Lines 18-19: Are the measured data relatively accurate and the simulated data reproduce them (what is written) or is it the predictions that are relatively

good (what I assume the authors want to write). Please correct. Also, try to be quantitative and replace "relatively" with quantitative measures.

Reply: The sentence was reformulated (lines 18-19 page 1) as below:

*"At the Rifle site, the coupled, hydrological-thermal-geophysical inversion approach well predicted the matric potential, temperature, and apparent resistivity with the Nash-Sutcliffe efficiency criterion greater than 0.92."*

Page 2, line 4: Perhaps "play an important role in controlling" instead of "control". Other factors control this partitioning as well.

Reply: We changed this sentence to "*For example, watershed moisture content and temperature are the main factors that control the partitioning of precipitation into evapotranspiration, infiltration and runoff*" (lines 3-4 page 2).

Page 2, line 5: Is ecosystem moisture an accepted term?

Reply: We changed to "*For ecosystem, moisture content and temperature conditions*" (line 6 page 2)

Page 2, line 10: Perhaps "spatio-temporal". Reconsider the term "native". Generally speaking, this sentence would be easier to follow by clarifying what are the differences in scales between the so-called native processes and ecosystem functioning.

Reply: We changed to "spatio-temporal". The term "native" was replaced by "local". This replacement should clarify the scale differences.

Page 2, line 12: I suggest to remove "direct measurements". Soil moisture is typically inferred from dielectric properties. This is not a direct measurement. Indeed, one can discuss if direct measurements exists altogether.

Reply: "direct measurements" was removed

Page 2, lines 21-24: Data and models are not the same! Hubbard et al. (2001) integrate geophysical models/tomograms with point measurements. On line 24, add "pumping tests" as these are needed to get the average hydraulic conductivity, while the flowmeter gives relative variations.

Reply: We restated the sentence (line 20-22 page 2) as below: *"Statistical approaches have been extensively used to integrate point measurements with commonly geophysical models/tomograms, such as Ground Penetrating Radar (GPR) and Electrical Resistance Tomography (ERT)."*

"pumping tests" was added in the revised version

Page 3, line 2: I think it should be "hillslope".

Reply: This was corrected

Page 3, lines 3-4: Unfortunately, few hydrogeophysical inversion approaches are developed to "improve quantification of subsurface processes". They are in most case only used for parameter estimation, which is something very different and much more restrictive.

Reply: We changed the "improve quantification of subsurface processes" to "estimate soil hydrological parameters". (lines 4-5 page 3)

Page 3, line 6: You need a qualifier before inversion, something like "hydrological".

Reply: We modified the sentence (lines 6-8 page 3) as below *"Because coupled inversion approach links hydrological simulation outputs (e.g., moisture content) with geophysical data, it avoids the errors typically associated with geophysical inversion process"*

Page 3, line 13: I don't think that "its" is used correctly here. Please rephrase.

Reply: We changed "its" to "their"

Page 3, line 24: "at high resolution (add references)".

Reply: We deleted the paragraph that contains this phase

Page 4, line 1: This is true for near-surface applications (please clarify). Seismological stations are probably used much more for geophysics in general. I would call ERT a method, not an approach.

We modified the sentence in the revised version (lines 21-22 page 3) as below: *"To date, ERT is the geophysical technique that is most commonly collected in an autonomous manner for near-surface applications"*

Page 4, line 12: Replace "and" with "which". Line 17, remove "the", line 18 change to "images". Line 26, replace "into a" with "within".

Reply: These were changed.

Page 5, line 7: I would remove references to Archie here. Archie defined the formation factor and the effect of saturation. However, referring to Archie implies that surface conductivity is ignored, which is not done here.

Reply: "Archie" was removed

Page 6, line 8: Change symbol for porosity to be consistent with the rest of the paper.

Reply: The symbol was changed

Page 6 and elsewhere: Equations are part of the sentences, so use "," and "." As appropriate after the equations. Ensure that variables are in italics. At the moment, they are sometimes, but not always (e.g., line 19).

Reply: Commas were added after the equations

Page 7, line 18: Perhaps extend this aspect in the discussion? Is this a big issue?

Reply: In the Rifle site study, transpiration and plant water uptake are not a big issue in this study because the study site is mostly bare soil. However, in the case that soil is largely covered by vegetation, vegetation should be considered. We are working with a land community model (CLM) to address this issue.

Page 8, line 11: Why not use subscript "b", as sigma_a or rho_a suggest apparent properties and not bulk properties.

Reply: We replaced subscript "a" by "b"

Page 8, lines 17-18: It is ok to fix the cementation index, but I am not sure if I see the argument here. The surface contribution is a constant over time while the saturation contribution is time-variable, so identifiability should not be an issue if enough variations in the states are probed.

Reply: Our sample measurements show that water conductivity does not much change over time and space. Because we had measurement value so we fixed in the inversion. We don't know surface conductance of the fill layer so we attempted to estimate it.

Page 9, lines 24-29: How did you balance the different types of data? Ideally, one should have a WRMSE of 1 for each data type after inversion and using actual data errors. In practice, this is essentially never done. The ERT data are given errors of 5% to somehow accommodate modeling/petrophysical errors, but they are presented as observational errors while the actual observational errors are probably less than 1%. I assume similar common "tricks" are used for the other data types. I don't criticize this, but some discussion about how the data were weighted and reporting the final data misfits for the individual data types should be part of the paper.

Reply:  We used measurement errors (inverse covariance matrix in equation 15) to account for different types. It is reasonable because in theory, it represents the likelihood function. In application, it implies that the more accurate observation will have stronger weight in the objective function. The "trick" here is that we did not know exact measurement errors, so we subjectively set measurement errors based on our evaluation of the accuracy of each measurement type.

Page 12, line 3: "at the within". Please revise.

Reply: This was corrected as below (lines 3-4 page 12): "The newly developed approach was tested at a floodplain adjoining the Colorado River, near Rifle, Colorado"

Page 13, line 13: It should be Günther. Also Rücker when his name is used.

Reply: This was corrected

Pager 13, lines 15-18. This is a bit of a circular argument (a similar construct is also found later in the paper). Why not write something like: As expected, the clay-rich fill and Wasatch layers show up as less resistive layers in the ERT : : :".

Reply: We reformulated these sentences to "As expected, the clay-rich fill and Wasatch layers exhibit less resistive than the alluvium layer" (line 8 page 13).

Page 13, line 18: Why not across the full ERT line?

Reply: In our project, we need to provide information on hydrological and thermal dynamics for other studies at a part of the transect around the TTO2 well. So to save the computational time, we performed inversion in this part of transect only.

Page 13, lines 21-22. Give references that support these values.

Reply: That is the average value estimated from our core drilling and sampling.

Page 13, line 23: This is ok for the hydrological model, but clearly insufficient for the ERT modeling. Express how you extend the ERT model and account for the Wasatch layer.

Reply: See our previous response.

Page 14, line 2: "the meteorological".

Reply: This word was corrected.

Page 14, line 13: Write "apparent resistivity".

Reply: This phrase was corrected.

Page 14, line 14: State what this implies in terms of assumptions. Using data from a well that is not located in the modeled study area.

Reply: We add the following assumption to the revised version (lines 27-28 page 14):

*Assuming that the lateral variation in subsurface temperature between TTO2 and TTO3 wells (see the TT03 location in Figure 2) was insignificant, we used the temperature data at the TTO3 well for inversion.*

Everywhere: Please consider to replace "heat conductivity" with the more common term: "thermal conductivity".

Reply: We replaced "heat conductivity" by "thermal conductivity"

Page 18, line 14: Replace "ignorable" with "negligible".
Reply: We replaced "ignorable" with "negligible".

Page 19, line 5: It should be "reproduce".

Reply: It was corrected

Page 19, line 27: Write "high apparent resistivity values", same for line 28. Also, on line 28: replace "located deeper" with "more sensitive to deeper locations".

Reply: These were corrected

Page 20, lines 1-5: This problem would not appear in a log-log plot and it would be more consistent with the error model.

Reply: In these sentences, we meant that because we used relative error to set the measurement error of ERT. As a result, the absolute error of higher apparent resistivity is larger than that of lower apparent resistivity. This leads to lower weight of high apparent resistivity in the objective function.

Page 20, line 21: It seems impossible that the downward flux is 24 mm/s. Please clarify what this value refers to? I interpret this in the same way as a Darcy flux and this seems unrealistically high.

Reply: We corrected this information. It it 24 mm/day.

Page 20, line 25: Replace "to the upward flow up to" with "upward flow starting at".

Reply: We replaced "to the upward flow up to" with "upward flow starting at".

Page 21, line 5: Add "estimated" before "subsurface".
Reply: We added "estimated"

Page 21, line 15: State how much noise was added.

Reply: We modified that sentence as below (lines 4-5 page 22):
*"Add Gaussian white noise (zero mean and standard deviation of 5% of artificial*

*apparent resistivity data) to the artificial data to obtain the synthetic data."*

Page 22, lines 22-24: This is a worthy contribution, but it seems as if not accounting for the variations of salinity is a bit limiting. The precipitation will clearly be much more resistive than the groundwater. Perhaps something to comment on?

Reply: We agree with the reviewer that considering the variation of salinity is necessary. It would be implemented in the next study. We added a sentence that discusses how to consider this variation in hydrogeophysical inversion in line 13-16 page 8.

Page 24, lines 1-4: Perhaps comment on the potential of filtering approaches/data assimilation in the context of this type of problems?

Reply: We add some filter techniques (Kalman ensemble filter, maximum likelihood ensemble filter, particle filter) that we will use in future research (line 19-20 page 24).

Figure 2: What are the East-West trending yellow dots. What are the green dots. Explain or remove.

Reply: These dots show the locations of other biochemical measurements. We removed them

Figure 3: The depth is positive with depth, so please remove "-" signs". Also give units and properties shown (log10 of resistivity). Add "inferred" before "fill-alluvium". Also, explain how this was done somewhere in the paper.

Reply: The figure was corrected. The explanation that shows how to determine the interfaces between layers was added to the revised version.

---

## Author Comment (AC2) · 20 Jul 2016

We are grateful Dr. Boaga for his valuable comments, corrections and suggestions. His review significantly helped us to improve the quality of our manuscript.

General Comments The paper concerns an interesting and innovative procedure of coupled inversion for hydrological and geophysical parameters. In particular Authors present double phases air-water-heat inversion coupled with ERT data. Tran et al. show the results for both synthetic test and real data, with a relevant sensitivity analysis of the parameter estimation which highlights state variables resolution capabilities. The topic is of interest for HESS readers and the manuscript is concise and well written. Despite this, I suggest minor revision before the acceptance for publication: some points need to be better clarified, some figures should be revised cause they are not at the level of the paper, some captions need improvement. Here below a list of detailed comments.

Specific Comments

The Manuscript comes without a continuing lines numbering, restarting in every pages. This usually does not help the revision work. I will refer to page number and relative line numbering.

Reply: We are sorry for the inconvenience of manuscript. We have to follow the format of HESS Discussion. We do hope this format will be modified in the near future to help reviewers to easily review manuscripts.

Pg3 Ln 20-25 Sentences not clear to me. You cannot assert that difficulties in hydrogeophysics are linked to temporal/spatial resolution variables problems, and then refer to the high resolution of autonomous acquisitions (?). As in the paper of Binley et al. you cited, several methods are applied second their proper potentials. On the other hand I agree with you we are still far in properties characterization, and your paper represents a relevant step forward.

Reply: After reviewing, we found that this paragraph is not much relevant to our study so we deleted it.

Pg4 Ln 8. I suggest suspension points after properties affecting resistivity description, cause it is not (and cannot be) exhaustive. Ln 24. I'm always prudent when reading there are no similar attempts in the previous literature. Some different works partially fronted the topic (e.g. Jardani et al 2013; Irving et . 2010), but this does not affect the quality of yours work. Probably the differences stay in the design of coupled hydro-geophysical inversion, and the level of results obtained.

Reply: We restated these sentences (line 28 page 3 and line 16 page 4)

Pg6 There is probably an error in the symbol of porosity in Eq2 or in the text.

Reply: The error was corrected (line 1 page 6)

Pg9 ln3. There's no need to introduce the relation between resistivity and conductivity here. Ln 10-15. (fig1). The flowchart is unclear and the caption confusing. I understand it is ambitious describe graphically all the inversion scheme, but I suggest to redrawn the flowchart and explain better this crucial part of your relevant work.

Reply: A sentence that describes the relationship between conductivity and resistivity was removed. A new hydrogeophysical inversion flowchart and caption were added to the revised version (page 31) and presented as below:

[Figure]

*Figure 1:* *Flowchart showing the steps involved in the coupled hydrological-thermal-geophysical inversion scheme. The objective function is represented by Equation (15). Estimated parameters consist of hydrological-thermal and petrophysical parameters (blue rectangles). The navy blue rectangles denote the model inputs, including prior information about estimated parameters, and the top and bottom boundary conditions. The purple rectangles denote the forward TOUGH2, geophysical and petrophysical models. The teal and indigo rectangles, respectively, denote the simulation and measurement. Data for inversion in this study include matric potential, subsurface temperature and apparent resistivity.*

Pg10.Ln20. I do not understand the sentence about initial guesses, it sounds unnecessary. Explain simply your (elegant) procedure.

Reply: This part is a practical procedure that I used in this study to step-by-step estimate different hydrological, thermal and petrophysical parameters. The optimization algorithm used in this study is a local optimization algorithm, while the inverse problem is

nonlinear. As a result, the hydrological-thermal-geophysical inversion can be trapped into local optimal solution if the initial guesses of estimated parameters are not good. We tackled this problem by:

- Use matric potential data to estimate hydrological parameters
- Fix the hydrological parameters that were obtained in step 1, use temperature data to estimate thermal parameters
- Use hydrological and thermal parameters obtained in step 1 and 2 as initial guesses for hydrological-thermal-geophysical inversion that estimates hydrological-thermal and petrophysical parameters using matric potential, temperature and apparent resistivity data

Pg12Ln5 (fig2). Map figure should be re-drawn, it is not well readable, missing coordinates, label and fonts are too small
Reply: A new figure was added to replace the old one (page 32) as below

39.531, -107.776                         39.531, -107.768

39.528, -107.776                         39.528, -107.768

**Figure 2:** Plan view of the Rifle floodplain of the Colorado River, Colorado, and the location of the TT02 and TT03 wells and ERT line.

Pg13.Ln1 What is the relationship with yours work and Arora et al. one? Not clear why do you present extensively this biogeochemical topic here. Ln 18. Why the hydrological model contains simply two geological layers? This is not specified. Ln 19 Please insert citation introducing Wasatch layers resistivity, otherwise these seems reasonable but not supported considerations (and should be placed in the discussion section). Fig.3 should be revised. Please put unit over the scale and not with X label. It is unusual the starting with negative distance values.

Reply: We presented these studies to show that quantification of hydrological and thermal dynamics in this study are crucial for other biochemical studies in our study site.

In our site study, the subsurface has two clear layers (fill and alluvium layers). Considering heterogeneity of soil properties inside each layer is possible but it also increases the number of estimated parameters, which may lead to non-uniqueness problem. As a result, we considered the subsurface include only two layers and estimate the average soil properties of each layer.

In this study the resistivity of Wasatch layer was obtained from ERT geophysical inversion, which is around $45 \ \Omega. m.$ (line 5 page 14).

Figure 3 was modified in the revised version (page 33) and presented as below. In this study, we determined that the computational domain of the hydrological simulation centered at the TTO2 well with a length of 30 m ($x_{TTO2}$=0). As a result, the first x-coordinate was x=-15 m and the end x-coordinate x=15 m.

[Figure]

**Figure 3:** *(a) The 2-D image of the soil electrical resistivity obtained by inverting ERT data collected on May 20, 2013. The magenta and white lines delineate the inferred fill-alluvium and alluvium-Wasatch boundaries. Green square markers denote the fill-alluvium boundary determined from the well logs of TT02 and TT03 and adjacent wells, as recorded in the field during drilling. The blue rectangular box indicates the hydrological-thermal computational domain. (b) Computational domain for the hydrological-thermal inversion with associated grid mesh. Blue and orange regions represent the fill and alluvium layers, respectively. The domain is situated below an atmospheric layer (top boundary) and above the relatively impermeable Wasatch (bottom boundary).*

Ln 23. Please introduce before the characteristics (e.g. a stratigraphy description) of TT02 and TT03 boreholes.

Reply: We presented the procedure to determine the soil stratigraphy using ERT geophysical inversion and TTO2 and TTO3 boreholes in lines 8-15 page 13 as below:

*To specify the locations of the fill-alluvium and alluvium-Wasatch interfaces from ERT geophysical inversion, we used the depths of these interfaces observed at the TTO2 and TTO3 well as the references to determine resistivity thresholds. Accordingly, a grid cell with a resistivity greater than 1.52 $Log_{10}(\Omega.m)$ and above 1.5 m depth belongs to the alluvium layer. The cells whose resistivity values are smaller than 1.83 $Log_{10}(\Omega.m)$ and below 5 m are assigned to the Wasatch layer. The remaining cells are in the fill layers. The magenta and white lines in Figure 3 represent the fill-alluvium and alluvium-Wasatch interfaces, respectively.*

Pg14 Ln 11. How do you approximate bottom temperature from land surface temperature?

Reply: It was extrapolated from measured temperature at above depths (z=4.6 and 6 m)

Fig.4 Caption should be improved to explain better these relevant sensitivity graphs.

Reply: We modified the Figure 4 caption as below (page 34):

*"The sensitivity coefficients |EE| of matric potential, subsurface temperature and apparent resistivity data with respect to different hydrological, thermal and petrophysical parameters. A parameter with a higher |EE| is more likely to be determined. (a) The sensitivity coefficient |EE| of the matric potential at depths of 0.5, 1, 1.5, 2, 2.5 and 3 m, with respect to the hydrological parameters of the fill and alluvium layers, and the gas diffusion coefficient standard conditions. (b) The |EE| of the temperature at depths of 0.75, 1, 1.5, 2.5, 4.63 and 6 m, with respect to the thermal conductivity of fill and alluvium layers. (c) The temporal variations of the |EE| of the resistivity data with respect to the soil hydrological-thermal and petrophysical parameters of both fill and alluvium layers."*

Tab.1 Table is quite confusing to me and caption does not help. Explain better what's from hydrological inversion and what's from the coupled one. Note: 'To' miss the initial bracket?

Reply: We modified the table caption to provide more information about inversion as below (page 40):

*Constraints and estimated values of the hydrological-thermal and petrophysical*

*parameters for different inversion cases. Hydrological inversion used matric potential data to estimate hydrological parameters (m (fill, alluvium), α (fill, alluvium), K (fill) and D). Thermal inversion used subsurface temperature data to estimate thermal conductivity of both fill and alluvium layers (λ (fill, alluvium)). Coupled inversion used all matric potential, temperature and apparent resistivity data to estimate parameters m (fill), α (fill), λ (fill), n (fill), n (alluvium) and σ (fill).*

Thanks for the reading of a very interesting and well written work, which in my opinion opens important further perspectives for hydrological characterization.